# EVALUATING ROBUSTNESS TO UNFORESEEN ADVERSARIAL ATTACKS

## ABSTRACT

When considering real-world adversarial settings, defenders are unlikely to have access to the full range of deployment-time adversaries during training, and adversaries are likely to use realistic adversarial distortions that will not be limited to small $L_p$-constrained perturbations. To narrow in on this discrepancy between research and reality we introduce eighteen novel adversarial attacks, which we use to create ImageNet-UA, a new benchmark for evaluating model robustness against a wide range of unforeseen adversaries. We make use of our benchmark to identify a range of defense strategies which can help overcome this generalization gap, finding a rich space of techniques which can improve unforeseen robustness. We hope the greater variety and realism of ImageNet-UA will make it a useful tool for those working on real-world worst-case robustness, enabling development of more robust defenses which can generalize beyond attacks seen during training.

## 1 INTRODUCTION

Neural networks perform well on a variety of tasks, yet can be consistently fooled by minor adversarial distortions (Szegedy et al., 2013; Goodfellow et al., 2014). This has led to an extensive and active area of research, mainly focused on the threat model of an "$L_p$-bounded adversary" that adds imperceptible distortions to model inputs to cause misclassification. However, this classic threat model may fail to fully capture many real-world concerns regarding worst-case robustness (Gilmer et al., 2018). Firstly, real-world worst-case distributions are likely to be varied, and are unlikely to be constrained to the $L_p$ ball. Secondly, developers will not have access to the worst-case inputs to which their systems will be exposed to. For example, online advertisers use perturbed pixels in ads to defeat ad blockers trained only on the previous generation of ads in an ever-escalating arms race (Tramèr et al., 2018). Furthermore, although research has shown that adversarial training can lead to overfitting, wherein robustness against one particular adversary does not generalize (Dai et al., 2022; Yu et al., 2021; Stutz et al., 2020; Tramer & Boneh, 2019), the existing literature is still focuses on defenses that train against the test-time attacks. This robustness to a train-test distribution shift has been studied when considering average-case corruptions (Hendrycks & Dietterich, 2018), but we take this to the worst-case setting.

We address the limitations of current adversarial robustness evaluations by providing a repository of nineteen gradient-based attacks, which are used to create ImageNet-UA—a benchmark for evaluating the *unforeseen robustness* of models on the popular ImageNet dataset (Deng et al., 2009). Defenses achieving high Unforeseen Adversarial Accuracy (UA2) on ImageNet-UA demonstrate the ability to generalize to a diverse set of adversaries not seen at train time, demonstrating a much more realistic threat model than the $L_p$ adversaries which are a focus of the literature.

Our results show that unforeseen robustness is distinct from existing robustness metrics, further highlighting the need for a new measure which better captures the generalization of defense methods. We use ImageNet-UA reveal that models with high $L_\infty$ attack robustness (the most ubiquitous measure of robustness in the literature) do not generalize well to new attacks, recommending $L_2$ as a stronger baseline. We further find that $L_p$ training can be improved on by alternative training processes, and suggest that the community focuses on training methods with better generalization behavior. Interestingly, unlike in the $L_p$ case, we find that progress on CV benchmarks has at least partially tracked unforeseen robustness. We are hopeful that ImageNet-UA can provide an improved progress measure for defenses aiming to achieve real-world worst-case robustness.

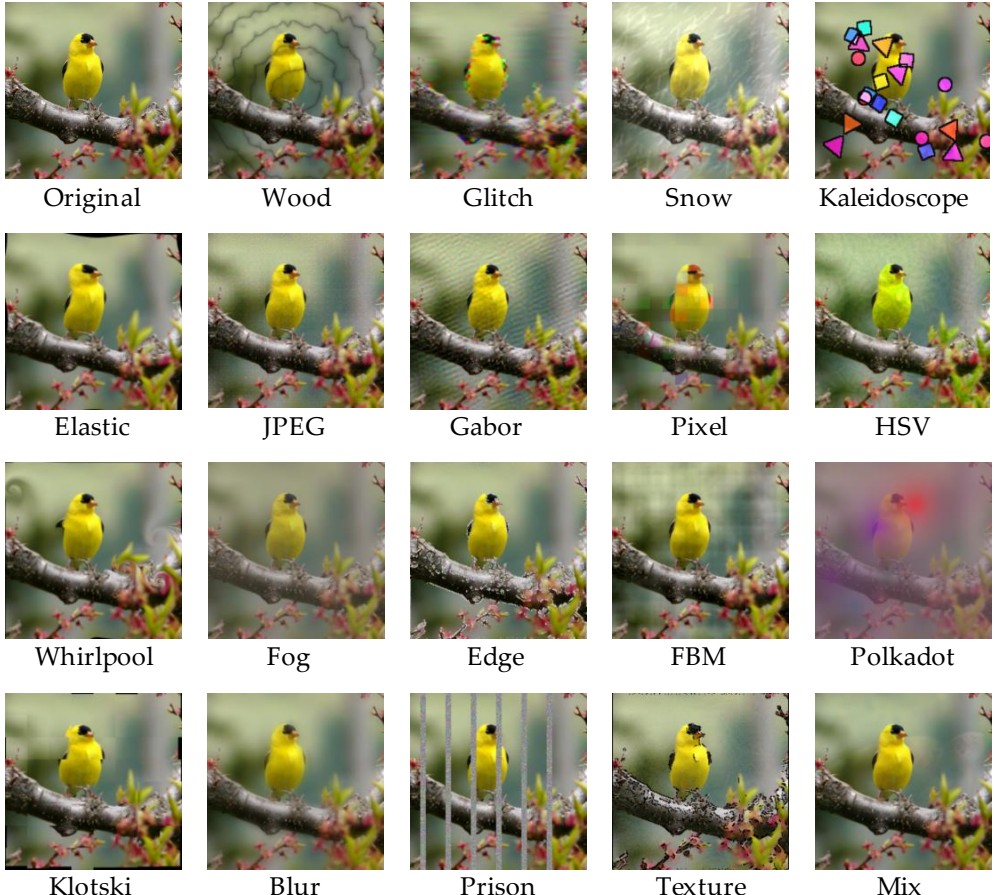

Figure 1: **The full suite of attacks.** We present nineteen differentiable non-$L_p$ attacks as part of our codebase, eighteen of which are novel. To aid visualization, we use higher distortion levels in this figure than in our benchmark. See Appendix G for examples of the distortion levels used within our benchmark, and Appendix K for a human study on semantic preservation.

To summarize, we make the following contributions:

- We design eighteen novel non-$L_p$ attacks, constituting a large increase in the set of dataset-agnostic non-$L_p$ attacks available in the literature. The full benchmark consists of the nineteen attacks shown in Figure 1. , which are split into a validation and test set.

- We make use of these attacks to form a new benchmark (ImageNet-UA), standardizing and greatly expanding the scope of unforeseen robustness evaluation.

- We show that it UA2 is distinct from existing robustness metrics in the literature, and demonstrates that classical $L_p$-training focused defense strategies can be improved on. We also measure the unforeseen robustness of a wide variety of techniques, finding promising research directions for generalizing adversarial robustness.

## 2 RELATED WORK

**Evaluating Adversarial Robustness.** Adversarial robustness is notoriously difficult to evaluate correctly (Papernot et al., 2017; Athalye et al., 2018). To this end, Carlini et al. (2019) provide extensive guidance for sound adversarial robustness evaluation. Our ImageNet-UA benchmark incorporates several of their recommendations, such as measuring attack success rates across several magnitudes of distortion and using a broader threat model with diverse differentiable attacks. Existing measures

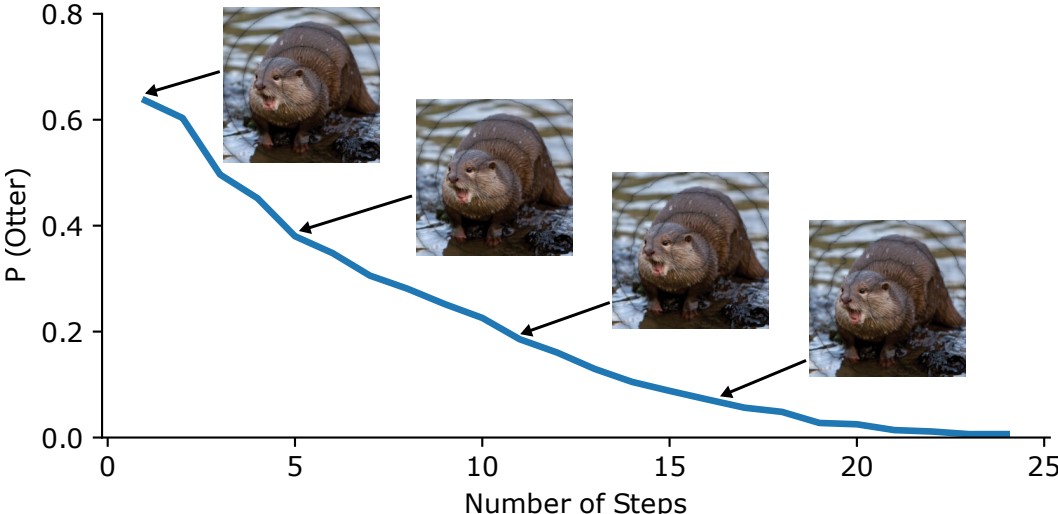

Figure 2: **Progression of an attack.** As we optimize our differentiable corruptions, model performance decreases, while leaving the image semantics unchanged. Unoptimized versions of our attacks have a moderate impact on classifier performance, similar to common corruptions (Hendrycks & Dietterich, 2019), while optimized versions cause large drops in accuracy.

of adversarial robustness (Croce & Hein, 2020; Moosavi-Dezfooli et al., 2015; Weng et al., 2018) almost exclusively, apply only to attacks optimizing over an $L_p$-ball, limiting their applicability for modeling robustness to new deployment-time adversaries.

**Non-$L_p$ Attacks.** Many attacks either use generative models (Song et al., 2018; Qiu et al., 2019) that are often hard to bound and are susceptible to instabilities, or make use of expensive brute-force search techniques Engstrom et al. (2017). We focus on attacks which are fast by virtue of differentiability, applicable to variety of datasets and independent of auxiliary generative models. Previous works presenting suitable attacks include Laidlaw & Feizi (2019); Shamsabadi et al. (2021); Zhao et al. (2019), who all transform the underlying color space of an image and Xiao et al. (2018) who differentiably warp images, and which we adapt to create our own Elastic attack. The literature does not have a sufficiently diverse set of suitable adversaries to effectively test the generalization properties of defenses, causing us to develop our suite of attacks.

**Unforeseen and Multi-attack Robustness.** There exist defense methods which seek to generalize across an adversarial train-test gap (Dai et al., 2022; Laidlaw et al., 2020; Lin et al., 2020). Yet, comparison between these methods is challenging due to the lack of a standardized benchmark and an insufficient range of adversaries to test against. We fill this gap by implementing a unified benchmark for testing unforeseen robustness. The more developed field of multi-attack robustness (Tramer & Boneh, 2019) aims to create models which are robust to a range of attacks, but works generally focus on a union of $L_p$ adversaries (Maini et al., 2020; Madaan et al., 2021a; Croce & Hein, 2022) and do not enforce that test time adversaries have to differ from those used during training.

**Common corruptions** Several of our attacks (Pixel, Snow, JPEG and Fog) were inspired by existing common corruptions (Hendrycks & Dietterich, 2018). We fundamentally change the generation methods to make these corruptions differentiable, allowing us to focus on worst-case robustness instead of the average-case robustness (see Section 5.1 for empirical an empirical comparison).

## 3 THE UNFORESEEN ROBUSTNESS THREAT MODEL

**Action Space of Adversaries.** The allowed action space of an adversary is defined using a *perturbation set $S_x$* of potential adversarial examples for each input $x$. Given such a set, and a classifier $f$ which correctly classifies a point $x$ with its ground truth label $y$, an *adversarial example $x_{\text{adv}}$* is defined to be a member the perturbation set $S_x$ which causes the classifier to give an incorrect prediction:

$$x_{\text{adv}} \in S_x : f(x_{\text{adv}}) \neq f(x) \tag{1}$$

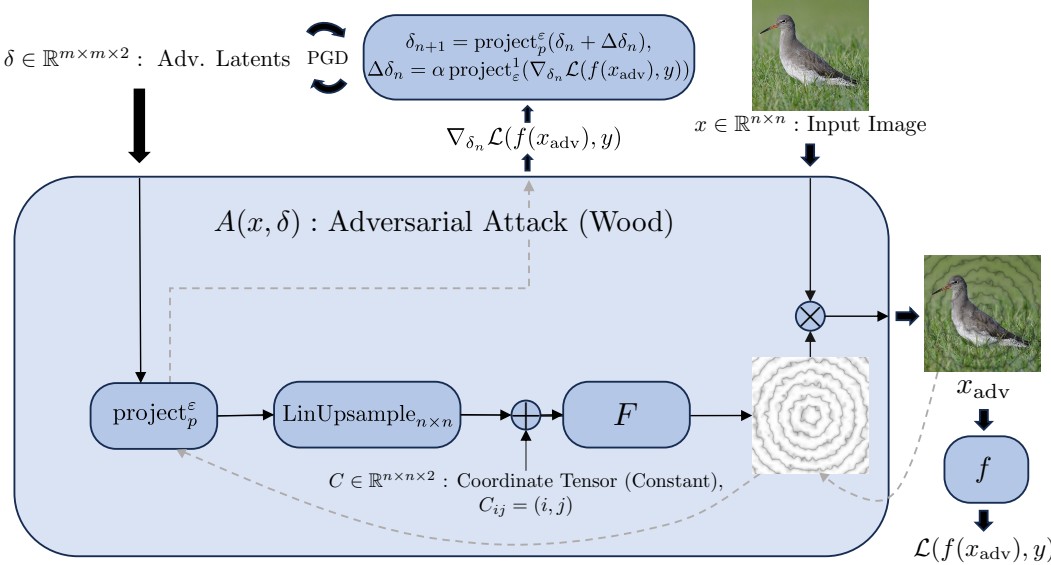

Figure 3: **An illustrative example of one of our attacks.** As demonstrated by this illustration of our Wood attack, all of our attacks function by performing PGD optimization on a set of latent variables. In the case of the Wood attack, these latent variables are inputs to concentric sine waves $(F(x,y) = \sin(\sqrt{x^2 + y^2}))$ which are overlaid on the image. See Appendix C for a more detailed explanation. We design effective attacks which are fast, easy to optimize, precisely bound, preserve image semantics, are portable across datasets and have variable intensity through the $\varepsilon$ parameter.

Then, under some distribution $\mathcal{D}$ of interest, the task of adversarial defenses is typically to achieve high accuracy in the face of an adversary which is allowed to optimize within the perturbation set.

We define the *unforeseen robustness* of a classifier as the accuracy of the classifier when faced with an unforeseen distribution of adversaries:

$$\mathbb{E}_{(x,y),A \sim \mathcal{D},\mathcal{A}} \left[ \min_{x_{\mathrm{adv}} \in S_x^A} \left\{ \mathbf{1}_{f(x_{\mathrm{adv}})=y} \right\} \right]$$

This is similar to the usual *adversarial accuracy* (Madry et al., 2017a), but instead of including a single $L_p$ adversary, we define a diverse distribution of adversaries $\mathcal{A}$ (where each adversary $A \in \mathbf{Dom}(\mathcal{A})$ defines a different perturbation set $S_x^A$ for each input $x$). Crucially, $\mathcal{A}$ is a uniform distribution over a held-out test set of adversaries. As we describe below, these adversaries cannot be used for training or hyperparameter tuning. We also provide a validation set of adversaries. Along with other attacks such as PGD, these attacks can be used for developing methods.

**Information Available to the Adversaries.** To ensure that our adversaries are as strong as possible (Carlini et al., 2019), and to avoid the usage of expensive black-box optimization techniques, we allow full white-box access to the victim models.

**Constraints on the Defender.** We enforce that defenders allow adversaries to compute gradients, in line with previous work demonstrating that defenses relying on masking of gradients are ineffective (Athalye et al., 2018). We also enforce that defenses do not make use of access to adversaries which are part of the test-time distribution $\mathcal{A}$. This assumption of unforeseen adversaries is contrary to most of the literature where the most powerful defenses involve explicitly training against the test time adversaries (Madry et al., 2017b), and allows us to model more realistic real-world situations where it is unlikely that defenders will have full knowledge of the adversaries at deployment time.

## 4 MEASURING UNFORESEEN ROBUSTNESS

To evaluate the unforeseen robustness of models, we introduce a new evaluation framework consisting of a benchmark ImageNet-UA and metric UA2 (Unforeseen Adversarial Accuracy). We also

Table 1: $L_p$ **robustness is disctinct from unforeseen robustness.** We highlight some of the models which achieve high UA2, while still being susceptible to $L_p$ attacks. Models below the dividing line are adversarially trained, with norm constraints in parentheses. These models demonstrate that unforeseen robustness is distinct from achieving $L_p$ robustness.

| Model | $L_\infty$ ($\varepsilon = 4/255$) | UA2 |
|---|---|---|
| Dinov2 Vit-large | 27.7 | **27.2** |
| Convnext-V2-large IN-1k+22K | 0.0 | 19.2 |
| Swin-Large ImageNet1K | 0.0 | 16.2 |
| ConvNext-Base $L_\infty$, ($\varepsilon = 8/255$) | **58.0** | 22.3 |
| Resnet-50, $L_\infty$ ($\varepsilon = 8/255$) | 38.9 | 10 |
| Resnet-50 $L_2$, ($\varepsilon = 5$) | 34.1 | 13.9 |

further release our nineteen (eighteen of which are novel) approaches for generating non-$L_p$ adversarial examples. We performed extensive sweeps to find the most effective hyperparameters for all of our attacks, the results of which can be found in Appendix A.

## 4.1 GENERATING ADVERSARIAL EXAMPLES

Each of our adversaries is defined by a differentiable function $A$ , which generates an adversarial input $x_\text{adv}$ from an input image $x$ and some latent variables $\delta$:

$$x_\text{adv} = A(x, \delta). \tag{2}$$

To control the strength of our adversary, we introduce an $L_p$ constraint to the variables $\delta$ (using $p = \infty$ or $p = 2$ ). We define our perturbation sets in terms of these allowed ranges of optimization variables, *i.e.*, for attack $A$ with epsilon constraint $\varepsilon$:

$$S_x^{A,\varepsilon} = \{A(x, \delta) \mid \|\delta\|_p \le \varepsilon\}.$$

As is typical in the literature (Madry et al., 2017b), we use our dataset loss function $\mathcal{L}$ to re-frame the finding of adversarial examples in our perturbation set Section 4.1 as a continuous optimisation problem, seeking $\delta_\text{adv}$ which solves:

$$\delta_\text{adv} = \underset{\delta:\|\delta\|_p \le \varepsilon}{\arg\min}\{\mathcal{L}(f(A(x, \delta)), y)\}, \tag{3}$$

and we then use the popular method of Projected Gradient Descent (PGD) (Madry et al., 2017b) to find an approximate solution to Equation (3).

Using this formulation helps us ensure that all of our attacks are independent of auxiliary generative models, add minimal overhead when compared to the popular PGD adversary (see Appendix E), are usable in a dataset-agnostic "plug-and-play" manner, can be used with existing optimization algorithms (see Figure 4a for behavior of attacks under optimization, and Figure 4a), come with a natural way of varying intensity through adjusting $\varepsilon$ parameter (see Figure 4b for behavior under varying $\varepsilon$), and have precisely defined perturbation sets which are not dependent on the solutions found to a relaxed constrained optimization problem. As discussed in Section 2, this is not the case for most existing attacks in the literature, prompting us to design our new attacks.

## 4.2 CORE ATTACKS

To provide fast evaluation, we select eight attacks to form our test set of attacks for unforeseen robustness. We refer to these as our core attacks, and we select them for diversity and effectiveness across model scale. We leave the other eleven attacks within our repository as a validation set for the tuning of defense hyperparameters and for a more complete evaluation of new techniques. We do not allow training or tuning on attacks that are visually similar to our core attacks (e.g., differentiable rain instead of snow). In Appendix D, we describe our process for designing these attacks and selecting the test and validation splits. The eight core attacks are:

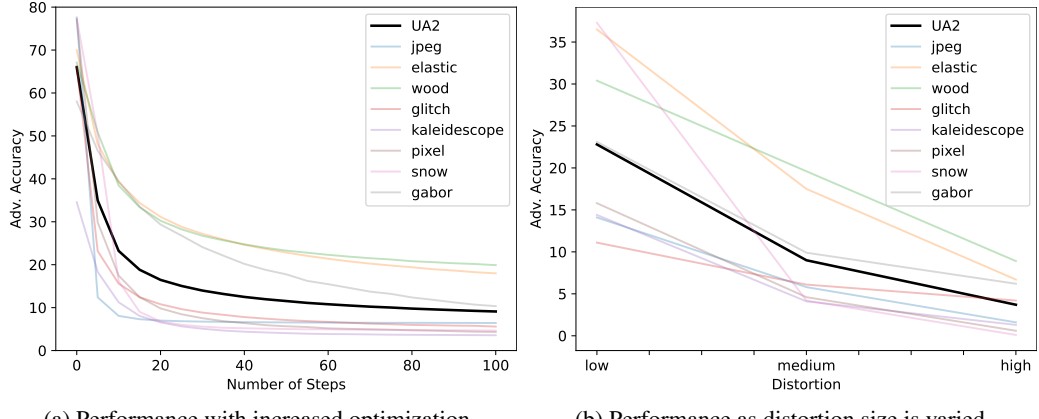

(a) Performance with increased optimization.  (b) Performance as distortion size is varied

Figure 4: **Attack effectiveness increases with optimization pressure and distortion budget.** We average performance against our core attacks across all our benchmarked models, demonstrating that our attacks respond to increased optimization pressure (Figure 4a). We further demonstrate the importance of the gradient-based nature by comparing random grid search to our gradient-based method in Appendix M. Furthermore, we demonstrate the ability for our attack stength to be customisable by showing that increasing distortion budget reduces model performance (Figure 4b).

**Wood.** The wood attack is described in Figure 3 and Appendix C.

**Glitch.** Glitch simulates a common behavior in corrupted images of colored fuzziness. Glitch greys out the image, splitting it into horizontal bars, before independently shifting color channels within each of these bars.

**JPEG.** The JPEG compression algorithm functions by encoding small image patches using the discrete cosine transform, and then quantizing the results. The attack functions by optimizing $L_\infty$-constrained perturbations within the JPEG-encoded space of compressed images and then reverse-transforming to obtain the image in pixel space, using ideas from Shin & Song (2017) to make this differentiable.

**Gabor.** Gabor spatially occludes the image with visually diverse Gabor noise (Lagae et al., 2009), optimizing the underlying sparse tensor which the Gabor kernels are applied to.

**Kaleidoscope.** Kaleidoscope overlays randomly colored polygons onto the image, and then optimizes both the homogeneous color of the inside of the shape, and the darkness/lightness of the individual pixels on the shape's border, up to an $L_\infty$ constraint.

**Pixel.** Pixel modifies an image so it appears to be of lower quality, by first splitting the image into $m \times m$ "pixels" and then and averaging the image color within each block. The optimization variables $\delta$ then control the level of pixelation, on a per-block bases.

**Elastic.** Our only non-novel attack. Elastic is adapted from (Xiao et al., 2018), functioning by which warping the image by distortions $x' = \text{Flow}(x, V)$, where $V : \{1, \dots, 224\}^2 \to \mathcal{R}^2$ is a vector field on pixel space, and Flow sets the value of pixel $(i, j)$ to the bilinearly interpolated original value at $(i, j) + V(i, j)$. To make the attack suitable for high-resolution images, we modify the original attack by passing a gaussian kernel over $V$.

**Snow.** Snow functions by optimising the intensity of individually snowflakes within an image, which are created by passing a convolutional filter over a sparsely populated tensor, and then optimising the non-zero entries in this tensor.

### 4.3 ImageNet-UA: A NEW BENCHMARK FOR UNFORESEEN ROBUSTNESS

We introduce ImageNet-UA, a benchmark for evaluating the unforeseen robustness of image classifiers on the popular ImageNet dataset (Deng et al., 2009). We also develop CIFAR-10 equivalent CIFAR-10-UA for computationally efficient evaluation of defense strategies and attack methods.

Table 2: ImageNet-UA **baselines** Here, we show some of the most robust models on ImageNet-UA, as well as baseline ResNet-50 models to compare between. We see a variety of techniques achieving high levels of robustness, demonstrating a rich space of possible interventions. The $L_\infty$ column tracks robustness against a PGD $L_\infty$ adversary with $\varepsilon = 4/255$. Numbers denote percentages.

| Model | Clean Acc. | $L_\infty$ | UA2 | JPEG | Elastic | Wood | Glitch | Kal. | Pixel | Snow | Gabor |
|---|---|---|---|---|---|---|---|---|---|---|---|
| DINOv2 ViT-large Patch14 | 86.1 | 15.3 | **27.7** | 14.3 | **42.6** | 39.7 | 17.7 | **46.2** | **17.2** | 14.2 | 29.9 |
| ConvNeXt-V2-large IN-1K+22K | **87.3** | 0.0 | 19.2 | 0.0 | 39.1 | 34.4 | 21.4 | 16.1 | 15.5 | 4.0 | 23.1 |
| ConvNeXt-V2-huge IN-1K | 86.3 | 0.0 | 17.7 | 0.0 | 42.5 | 21.2 | **23.8** | 24.3 | 6.6 | 0.7 | 22.2 |
| ConvNeXt-base, $L_\infty$ (4/255) | 76.1 | **58.0** | 22.3 | 39.0 | 23.8 | **47.9** | 12.9 | 2.5 | 9.7 | **30.2** | 12.8 |
| ViT-base Patch16, $L_\infty$ (4/255) | 76.8 | 57.1 | 25.8 | **52.6** | 26.3 | 47.2 | 13.8 | 8.1 | 11.9 | 27.1 | 19.5 |
| Swin-base IN-1K | 85.3 | 0.0 | 15.2 | 0.0 | 31.4 | 24.6 | 16.2 | 6.0 | 6.9 | 4.3 | **32.0** |
| ResNet-50 | 76.1 | 0.0 | 1.6 | 0.0 | 4.4 | 6.3 | 0.4 | 0.0 | 0.3 | 0.1 | 0.9 |
| ResNet-50 + CutMix | 78.6 | 0.5 | 6.1 | 0.2 | 17.9 | 15.5 | 2.5 | 0.1 | 6.7 | 3.0 | 2.7 |
| ResNet-50, $L_\infty$ (8/255) | 54.5 | 38.9 | 10.0 | 6.9 | 11.8 | 23.9 | 14.4 | 0.7 | 5.2 | 15.6 | 1.2 |
| ResNet-50, $L_2$ (5) | 56.1 | 34.1 | 13.9 | 39.7 | 11.9 | 19.4 | 12.2 | 0.3 | 9.7 | 15.4 | 2.5 |

Table 3: $L_p$ **training.** We train a range of ResNet-50 models against $L_p$ adversaries on ImageNet-UA

| Training | Train $\varepsilon$ | Clean Acc. | UA2 |
|---|---|---|---|
| Standard | - | **76.1** | 1.6 |
| $L_2$ | 1 | 69.1 | 6.4 |
| | 3 | 62.8 | 12.2 |
| | 5 | 56.1 | **13.9** |
| $L_\infty$ | 2/255 | 69.1 | 6.4 |
| | 4/255 | 63.9 | 7.9 |
| | 8/255 | 54.5 | 10.0 |

Table 4: $L_p$ **training on generated data.** We see the effect of training when training WRN-28-10 networks on CIFAR-10-50M, a 1000x larger diffusion-model generated version of CIFAR-10 (Wang et al., 2023)

| Dataset | Training | Clean Acc. | UA2 |
|---|---|---|---|
| CIFAR-10 | $L_2, \varepsilon = 1$ | 82.3 | 45.8 |
| | $L_\infty, \varepsilon = 8/255$ | 86.1 | 41.5 |
| CIFAR-10-50M | $L_2, \varepsilon = 0.5$ | 95.2 | **51.2** |
| | $L_\infty, \varepsilon = 4/255$ | 92.4 | **51.5** |

The unforeseen robustness achieved by a defense is quantified using a new metric, Unforeseen Adversarial Accuracy (UA2), which measures the robustness of a given classifier $f$ across a diverse range of unforeseen attacks. In line with Section 3 we model the deployment-time population of adversaries $\mathcal{A}$ as a categorical distribution over some finite set $\mathbf{A}$, with a distortion level $\epsilon_A$ for each adversary $A \in \mathbf{A}$. Section 3 then reduces to:

$$\text{UA2} := \frac{1}{|\mathbf{A}|} \sum_{A \in \mathbf{A}} \text{Acc}(A, \epsilon_A, f)$$

where $\text{Acc}(A, \varepsilon_a, f)$ denotes the adversarial accuracy of classifier $f$ against attack $A$ at distortion level $\varepsilon_A$. We select the population of adversaries to be the eight core adversaries from Section 4.2, setting $\mathbf{A} = \{$JPEG, Elastic, Wood, Glitch, Kaleidoscope, Pixel, Snow, Gabor$\}$.

We further divide our benchmark by picking three different distortion levels for each attack, leading to three different measures of unforeseen robustness: $\text{UA2}_{\text{low}}$, $\text{UA2}_{\text{med}}$ and $\text{UA2}_{\text{high}}$ (see Appendix A for specific $\varepsilon$ values used within this work), and we focus on focus on $\text{UA2}_{\text{med}}$ for all of our reports, referring to this distortion level as simply UA2. As distortion levels increase, model performance decreases (Figure 4b). We perform a human study (Appendix K) to ensure $\text{UA2}_{\text{med}}$ preserves image semantics.

## 5 BENCHMARKING FOR UNFORESEEN ADVERSARIAL ROBUSTNESS

In this section, we evaluate a range of models on our standardized benchmarks ImageNet-UA and CIFAR-10-UA. We aim to present a set of directions for future work, by comparing a wide range of methods. We also hope to explore how the problem of unforeseen robustness different from existing robustness metrics.

Table 5: **PixMix and $L_p$ training.** We compare UA2 performance on CIFAR-10 of models trained with PixMix and adversarial training. Combining PixMix with adversarial training results in large improvements in UA2, demonstrating the potential for novel methods to improve UA2. All numbers denote percentages, and $L_\infty$ training was performed with the TRADES algorithm.

| Training Strategy | Train $\varepsilon$ | Clean Acc. | UA2 |
|---|---|---|---|
| PixMix | - | **95.1** | 15.00 |
| $L_\infty$ | 4/255 | 89.3 | 37.3 |
| $L_\infty$ + PixMix | 4/255 | 91.4 | **45.1** |
| $L_\infty$ | 8/255 | 84.3 | 41.4 |
| $L_\infty$ + PixMix | 8/255 | 87.1 | **47.4** |

### 5.1 HOW DO EXISTING ROBUSTNESS MEASURES RELATE TO UNFORESEEN ROBUSTNESS?

We find the difference between existing popular metrics and UA2, highlighting the differential progress made possible by UA2:

**UA2 is distinct from existing measures of distribution shift.** We compare UA2 to the standard distribution-shift benchmarks given by ImageNet-C (Hendrycks & Dietterich, 2019), ImageNet-R Hendrycks et al. (2021) and ImageNet-Sketch (Wang et al., 2019). As shown in Appendix I, we find that performance on these benchmark correlates with non-optimized versions of our attacks. However, the optimized versions of our attacks have model robustness profiles more similar to $L_p$ adversaries. This highlights that UA2 is a measure of worst case robustness, similar to $L_p$ robustness, and distinct from other distribution shift benchmarks in the literature.

**$L_p$ robustness is correlated, but distinct from, unforeseen robustness.** As shown in Appendix L, unforeseen robustness is correlated with $L_p$ robustness. Our attacks also show similar properties to $L_p$ counterparts, such as the ability for black-box transfer (Appendix N). However, many models show susceptibility to $L_p$ adversaries while still performing well on UA2 (Table 1), and a range of strategies beat $L_p$ training baselines Section 5.2 . We conclude that UA2 is distinct from $L_p$ robustness, and present UA2 as an improved progress measure when working towards real-world worst-case robustness.

**$L_2$-based adversarial training outperforms $L_\infty$-based adversarial training** We see that $L_p$ adversarial training increases the unforeseen robustness of tested models, with $L_2$ adversarial training providing the largest increase in UA2 over standard training ($1.6\% \rightarrow 13.9\%$), beating models which are trained against $L_\infty$ adversaries ($1.6\% \rightarrow 10.0\%$). We present $L_2$ trained models as a strong baseline for unforeseen robustness, noting that the discrepancy between $L_\infty$ and $L_2$ training is particularly relevant as $L_\infty$ robustness is the most ubiquitous measure of adversarial robustness.

### 5.2 HOW CAN WE IMPROVE UNFORESEEN ROBUSTNESS?

We find several promising directions that improve over $L_p$ training, and suggest that the community should focus more on techniques which we demonstrate to have better generalization properties:

**Combining image augmentations and $L_\infty$ training.** We combine PixMix and $L_\infty$ training, finding that this greatly improves unforeseen robustness over either approach alone ($37.3 \rightarrow 45.1$, see Table 5). This is a novel training strategy which beats strong baselines by combining two distinct robustness techniques ($L_p$ adversarial training and data augmentation). The surprising effectiveness of this simple method highlights how unforeseen robustness may foster the development of new methods.

**Multi-attack robustness.** To evaluate how existing work on robustness to a union of $L_p$ balls may improve unforeseen robustness, we use CIFAR-10-UA to evaluate a strong multi-attack robustness baseline by (Madaan et al., 2021b), which trains a Meta Noise Generator (MNG) that learns the optimal training perturbations to achieve robustness to a union of $L_p$ adversaries. For WRN-28-10 models on CIFAR-10-UA, we see a large increase in unforeseen robustness compared to the best $L_p$ baseline ($21.4\% \rightarrow 51.1\%$, full results in Appendix J ), leaving scaling of such methods to full ImageNet-UA for future work.

Table 6: **Effects of data augmentation on UA2.** We evaluate the UA2 of a range of data-augmented ResNet50 models.

| Training | Clean Acc. | UA2 |
|---|---|---|
| Standard | 76.1 | 1.0 |
| Moex | **79.1** | **6.0** |
| CutMix | 78.6 | **6.0** |
| Deepaugment + Augmix | 75.8 | 1.8 |

Table 7: **Effects of pretraining and regularization on UA2.**

| Model | Clean Acc. | UA2 |
|---|---|---|
| ConvNeXt-V2-28.6M | **83.0** | **9.8** |
| ConvNeXt-V1-28M | 82.1 | 5.1 |
| ConvNeXt-V2-89M | **84.9** | **14.9** |
| ConvNeXt-V1-89M | 83.8 | 9.7 |
| ConvNeXt-V2-198M | **85.8** | **19.1** |
| ConvNeXt-V1-198M | 84.3 | 10.6 |

**Bounding perturbations with perceptual distance.** We evaluate the UA2 of models trained with Perceptual Adversarial Training (PAT) (Laidlaw et al., 2020). PAT functions by training a model against an adversary bounded by an estimate of the human perceptual distance, computing the estimate by using the hidden states of an image classifier. For computational reasons we train and evaluate ResNet-50s on a 100-image subset of ImageNet-UA, where this technique outperforms the best $L_p$ trained baselines ($22.6 \rightarrow 26.2$, full results in Appendix J).

**Regularizing high-level features.** We evaluate Variational Regularization (VR) (Dai et al., 2022), which adds a penalty term to the loss function for variance in higher level features. We find that the largest gains in unforeseen robustness come from combining VR with PAT, improving over standard PAT ($26.2 \rightarrow 29.5$, on a 100 class subset of ImageNet-UA, full results in Appendix J).

### 5.3 How has progress on CV benchmarks tracked unforeseen robustness?

**Computer vision progress has partially tracked unforeseen robustness.** Comparing the UA2 of ResNet-50 to ConvNeXt-V2-huge ($1\% \rightarrow 19.1\%$ UA2) demonstrates the effects of almost a decade of CV advances, including self-supervised pretraining, hardware improvements, data augmentations, architectural changes and new regularization techniques. More generally, we find a range of modern architectures and training strategies doing well (see Table 2, full results in Figure 7). This is gives a positive view of how progress on standard CV benchmarks has tracked underlying robustness metrics, contrasting with classical $L_p$ adversarial robustness where standard training techniques have little effect (Madry et al., 2017a).

**Scale, data augmentation and pretraining successfully improve robustness.** We do a more careful analysis of how three of the most effective CV techniques have improved robustness. As shown in Table 6, we find that data augmentation improves on unforeseen robustness, even in cases where they reduce standard accuracy. We compare the performance of ConvNeXt-V1 and ConvNeXt-V2 models, which differ through the introduction of self-supervised pretraining and a new normalization layer. When controlling for model capacity these methods demonstrate large increase unforeseen robustness Table 7.

## 6 Conclusion

In this paper, we introduced a new benchmark for *unforeseen adversaries* (ImageNet-UA) laying groundwork for future research in improving real world adversarial robustness. We provide nineteen (eighteen novel) non-$L_p$ attacks as part of our repository, using these to construct a new metric UA2 (Unforeseen Adversarial Accuracy). We then make use use this standardized benchmark to evaluate classical $L_p$ training techniques, showing that the common practice of $L_\infty$ training and evaluation may be misleading, as $L_2$ shows higher unforeseen robustness. We additionally demonstrate that a variety of interventions outside of $L_p$ adversarial training can improve unforeseen robustness, both through existing techniques in the CV literature and through specialised training strategies. We hope that the ImageNet-UA robustness framework will help guide adversarial robustness research, such that we continue making meaningful progress towards making machine learning safer for use in real-world systems.

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

## A  HYPERPARAMETERS

### A.1  TRAINED MODELS

To run our evaluations, we train a range of our own models to benchmark with:

- CIFAR-10 WRN-28-10 robust models and TRADES models are respectively trained with the official code of Rice et al. (2020) and Zhang et al. (2019) with the default hyperparameters settings
- The PAT-VR models on ImageNet100 were trained using the official code from Dai et al. (2022) and employed the hyperparameter settings outlined in the code of Laidlaw et al. (2020).
- ImageNet100 DINOv2 Oquab et al. (2023) models are trained by finetuning a linear classification head on the ImageNet100 dataset. We used a SGD optimizer with learning rate of 0.001 and employed early-stopping.

### A.2  MODEL REFERENCE

We use a range of baseline models provided by other works, with model weights available as part of their open source distribution:

- **ImageNet**
  - ConvNeXt models are from Liu et al. (2022)
  - ConvNeXt-V2 models are from Woo et al. (2023)
  - ViT models are from Steiner et al. (2022)
  - Swin models are from Liu et al. (2021)
  - Reversible-ViT models are from Mangalam et al. (2022)
  - CLIP (ViT-L/14) is from Radford et al. (2021)
  - DINOv2 models are from Oquab et al. (2023)
  - MAE models are from He et al. (2022)
- **CIFAR-10**
  - WideResNet TRADES models are from Zhang et al. (2019)
  - WRN + Diffusion models are from Wang et al. (2023)
  - Meta noise models are from Madaan et al. (2021b)
  - ResNet50 VR models are from Dai et al. (2022)
  - ReColorAdv models are from Laidlaw & Feizi (2019)
  - StAdv modesl are from Xiao et al. (2018)
  - Multi attack models are from Tramèr et al. (2018)
  - The Multi steepest descent model is from Maini et al. (2020)
  - PAT models are from Laidlaw et al. (2020)
  - Pre-trained ResNet18 $L_\infty$, $L_2$ and $L_1$ models are from Croce & Hein (2022)
- **ImageNet100**
  - ResNet50 PAT models are from Laidlaw et al. (2020)
  - ResNet50 PAT + VR models are from Dai et al. (2022)
  - DINOv2 models are from Oquab et al. (2023)

## A.3 ATTACK PARAMETERS

To ensure that our attacks are maximally effective, we perform extensive hyper-parameter sweeps to find the most effective step sizes.

Table 8: Attack parameters for ImageNet-UA

|  |  | Step Size | Num Steps | Low Distortion | Medium Distortion | High Distortion | Distance Metric |
|---|---|---|---|---|---|---|---|
| Core Attacks | PGD | 0.004 | 50 | 2/255 | 4/255 | 8/255 | $L_\infty$ |
|  | Gabor | 0.0025 | 100 | 0.02 | 0.04 | 0.06 | $L_\infty$ |
|  | Snow | 0.1 | 100 | 10 | 15 | 25 | $L_2$ |
|  | Pixel | 1 | 100 | 3 | 5 | 10 | $L_2$ |
|  | JPEG | 0.0024 | 80 | 1/255 | 3/255 | 6/255 | $L_\infty$ |
|  | Elastic | 0.003 | 100 | 0.1 | 0.25 | 0.5 | $L_2$ |
|  | Wood | 0.005 | 80 | 0.03 | 0.05 | 0.1 | $L_\infty$ |
|  | Glitch | 0.005 | 90 | 0.03 | 0.05 | 0.07 | $L_\infty$ |
|  | Kaleidoscope | 0.005 | 90 | 0.05 | 0.1 | 0.15 | $L_\infty$ |
| Extra Attacks | Edge | 0.02 | 60 | 0.03 | 0.1 | 0.3 | $L_\infty$ |
|  | FBM | 0.006 | 30 | 0.03 | 0.06 | 0.3 | $L_\infty$ |
|  | Fog | 0.05 | 80 | 0.3 | 0.5 | 0.7 | $L_\infty$ |
|  | HSV | 0.012 | 50 | 0.01 | 0.03 | 0.05 | $L_\infty$ |
|  | Klotski | 0.01 | 50 | 0.03 | 0.1 | 0.2 | $L_\infty$ |
|  | Mix | 1.0 | 70 | 5 | 10 | 40 | $L_2$ |
|  | Pokadot | 0.3 | 70 | 1 | 3 | 5 | $L_2$ |
|  | Prison | 0.0015 | 30 | 0.01 | 0.03 | 0.1 | $L_\infty$ |
|  | Blur | 0.03 | 40 | 0.1 | 0.3 | 0.6 | $L_\infty$ |
|  | Texture | 0.00075 | 80 | 0.01 | 0.03 | 0.2 | $L_\infty$ |
|  | Whirlpool | 4.0 | 40 | 10 | 40 | 100 | $L_2$ |

Table 9: Attack parameters for CIFAR-10-UA

|  |  | Step Size | Num Steps | Low Distortion | Medium Distortion | High Distortion | Distance Metric |
|---|---|---|---|---|---|---|---|
| Core Attacks | PGD | 0.008 | 50 | 2/255 | 4/255 | 8/255 | $L_\infty$ |
|  | Gabor | 0.0025 | 80 | 0.02 | 0.03 | 0.04 | $L_\infty$ |
|  | Snow | 0.2 | 20 | 3 | 4 | 5 | $L_2$ |
|  | Pixel | 1.0 | 60 | 1 | 5 | 10 | $L_2$ |
|  | JPEG | 0.0024 | 50 | 1/255 | 3/255 | 6/255 | $L_\infty$ |
|  | Elastic | 0.006 | 30 | 0.1 | 0.25 | 0.5 | $L_2$ |
|  | Wood | 0.000625 | 70 | 0.03 | 0.05 | 0.1 | $L_\infty$ |
|  | Glitch | 0.0025 | 60 | 0.03 | 0.05 | 0.1 | $L_\infty$ |
|  | Kaleidoscope | 0.005 | 30 | 0.05 | 0.1 | 0.15 | $L_\infty$ |
| Extra Attacks | Edge | 0.02 | 60 | 0.03 | 0.1 | 0.3 | $L_\infty$ |
|  | FBM | 0.006 | 30 | 0.02 | 0.04 | 0.08 | $L_\infty$ |
|  | Fog | 0.05 | 40 | 0.3 | 0.4 | 0.5 | $L_\infty$ |
|  | HSV | 0.003 | 20 | 0.01 | 0.02 | 0.03 | $L_\infty$ |
|  | Klotski | 0.005 | 50 | 0.03 | 0.05 | 0.1 | $L_\infty$ |
|  | Mix | 0.5 | 30 | 1 | 5 | 10 | $L_2$ |
|  | Pokadot | 0.3 | 40 | 1 | 2 | 3 | $L_2$ |
|  | Prison | 0.0015 | 20 | 0.01 | 0.03 | 0.1 | $L_\infty$ |
|  | Blur | 0.015 | 20 | 0.1 | 0.3 | 0.6 | $L_\infty$ |
|  | Texture | 0.003 | 30 | 0.01 | 0.1 | 0.2 | $L_\infty$ |
|  | Whirlpool | 16.0 | 50 | 20 | 100 | 200 | $L_2$ |

## B  DESCRIPTIONS OF THE 11 ADDITIONAL ATTACKS.

**Blur.**  Blur approximates real-world motion blur effects by passing a Gaussian filter over the original image and then does a pixel-wise linear interpolation between the blurred version and the original, with the optimisation variables controlling the level of interpolation. We also apply a Gaussian filter to the grid of optimisation variables, to enforce some continuity in the strength of the blur between adjacent pixels. This method is distinct from, but related to other blurring attacks in the literature (Guo et al., 2020; 2021).

**Edge.**  This attack functions by applying a Canny Edge Detector (Canny, 1986) over the image to locate pixels at the edge of objects, and then applies a standard PGD attack to the identified edge pixels.

**Fractional Brownian Motion (FBM).**  FBM overlays several layers of Perlin noise (Perlin, 2005) at different frequencies, creating a distinctive noise pattern. The underlying gradient vectors which generate each instance of the Perlin noise are then optimised by the attack.

**Fog.**  Fog simulates worst-case weather conditions, creating fog-like occlusions by adversarially optimizing parameters in the diamond-square algorithm (Fournier et al., 1982) typically used to render stochastic fog effects.

**HSV.**  This attack transforms the image into the HSV color space, and then optimises PGD in that latent space. Due to improving optimisation properties, a gaussian filter is passed over the image.

**Klotski.**  The Klotski attack works by splitting the image into blocks, and applying a differentiable translation to each block, which is then optimised.

**Mix.**  The Mix attack functions by performing differntiable pixel-wise interpolation between the original image and an image of a different class. The level of interpolation at each pixel is optimised, and a gaussian filter is passed over the pixel interpolation matrix to ensure that the interpolation is locally smooth.

**Polkadot.**  Polkadot randomly selects points on the image to be the centers of a randomly coloured circle, and then optimising the size of these circles in a differentiable manner.

**Prison.**  Prison places grey "prison bars" across the image, optimising only the images within the prison bars. This attack is inspired by previous "patch" attacks (Brown et al., 2017), while ensuring that only the prison bars are optimised.

**Texture.**  Texture works by removing texture information within an images, passing a Canny Edge Detector (Canny, 1986) over the image to find all the pixels which are at the edges of objects, and then filling these pixels in black—creating a silhouette of the original image. The other non-edge (or "texture") pixels are then whitened, losing the textural information of the image while preserving the shape. Per-pixel optimisation variable control the level of whitening.

**Whirlpool.**  Whirlpool translates individual pixels in the image by a differentiable function creating a whirlpool-like warpings of the image, optimising the strength of each individual whirlpool.

## C    FULL DESCRIPTION OF WOOD ATTACK

In Figure 3, we give a high-level explanation of the Wood attack. Here, we give a more detailed explanation of this figure.

Given a classifier $f$, the Wood attack with distortion level $\varepsilon$ functions by taking a set of adversarial latent variables $\delta_n \in \mathbb{R}^{m \times m \times 2}$ (representing a vector field of per-pixel displacements), applies $project_p^\varepsilon$ to project this field into the $\varepsilon$ ball in the $L_p$ metric and then uses bi-linear interpolation to upsample the latent variables to the input image size. The upsampled latent variables are then used to make the wood noise, by using an element-wise mapping $F \colon \mathbb{R}^{n \times n \times 2} \to \mathbb{R}^{n \times n}$, taking a coordinate to the (power of) the sine of its distance from the center of the image i.e. $F(I) = \sin(\sqrt{(X)^2 + (Y)^2})^\beta$, where $X_{ij} = I_{ij0} - n/2$ and $Y_{ij} = I_{ij1} - n/2$ and $\beta$ is an attack hyperparameter. When applied to constant coordinate tensor $C \in \mathbb{R}^{n \times n \times 2}$, $C_{ij} = (i, j)$, this function creates the distinctive "wood rings" of the Wood attack, which are then multiplied with the input image to produce adversarial input. By virtue of the differentiability of this process, we can backpropagate through this noise generation and optimize the adversarial image $x_{\text{adv}}$ by performing PGD (Madry et al., 2017a) on the input latent variables.

## D    PROCESS FOR DESIGNING ATTACKS AND SELECTING CORE ATTACKS

Our design of attacks is guided by two motivations: defending against unforeseen adversaries and robustness to long-tail scenarios. Unforeseen adversaries could implement novel attacks to, e.g., evade automated neural network content filters. To model unforeseen adversaries that might realistically appear in these scenarios, we include digital corruptions similar to what one might see on YouTube videos trying to evade content filters. These include attacks such as Kaleidoscope and Prison. To model long-tail scenarios, we include worst-case versions of common corruptions, like JPEG, Snow, and Fog.

In preliminary experiments, we found that some of these attacks were more effective than others, leading to lower accuracy with fewer steps. We also found that some attacks were more correlated with existing adversaries like PGD. For example, Prison is a pixel-level attack, so its results are more correlated with PGD than the other attacks. To increase the diversity and efficiency of our evaluation, we selected a core set of eight attacks based on their effectiveness and diversity, considering both visual diversity and the correlation of their results to each other and to PGD. This was an iterative process that led us to make substantial changes to some attacks. For example, we modified the implementation of the Elastic attack to use larger, lower-frequency distortions, which maintained its effectiveness while reducing correlation with PGD.

## E    ATTACK COMPUTATION TIME

We investigate the execution times of our attacks, finding that most attacks are not significantly slower than an equivalent PGD adversary.

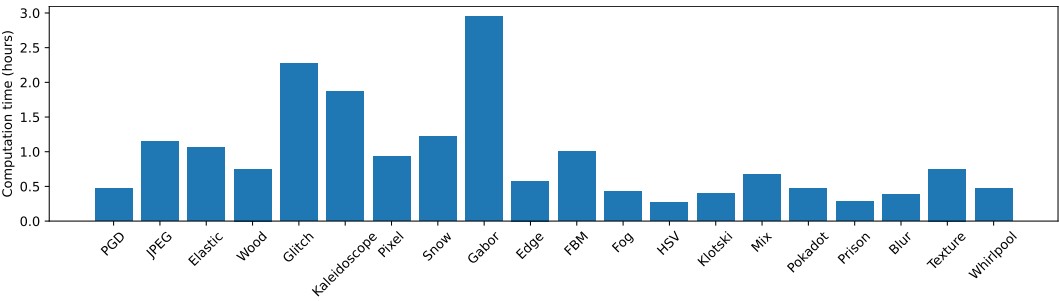

Figure 5: Evaluation time of the attacks on the ImageNet test set using a ResNet50 model with batch size of 200 on a single A100-80GB GPU, Attack hyper-parameters are as described in Appendix A.

# F  FULL RESULTS OF MODEL EVALUATIONS

We benchmark a large variety of models on our dataset, finding a rich space of interventions affecting unforeseen robustness.

## F.1  IMAGENET

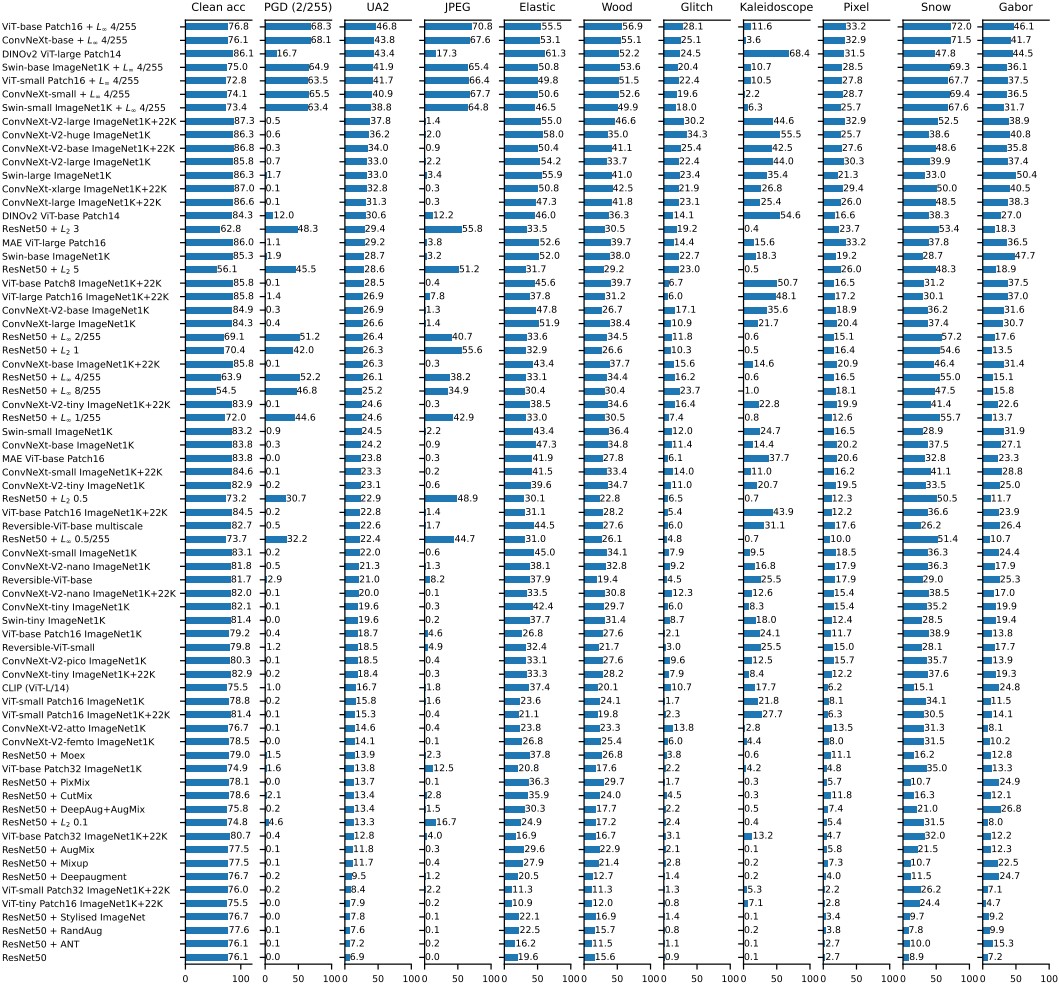

Figure 6: ImageNet UA2 performance under low distortion.

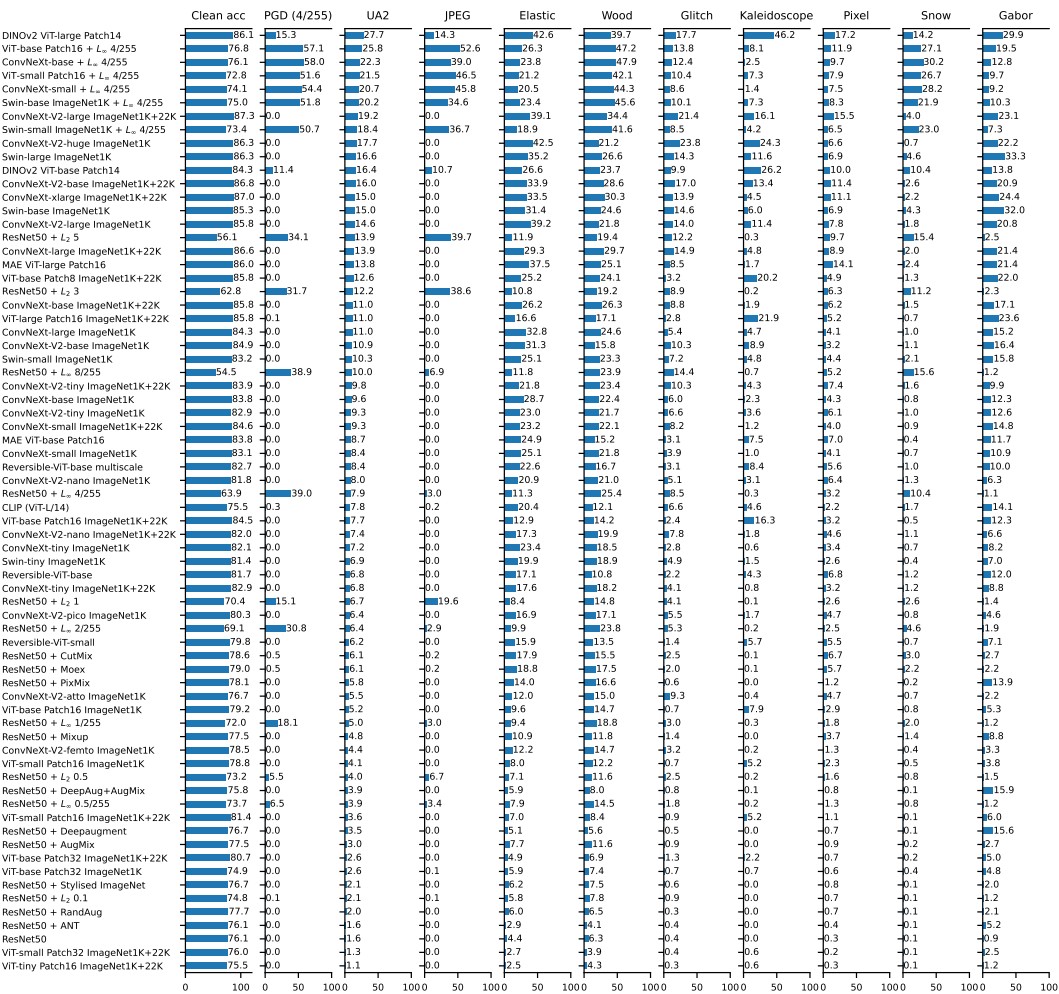

Figure 7: ImageNet UA2 performance under medium distortion

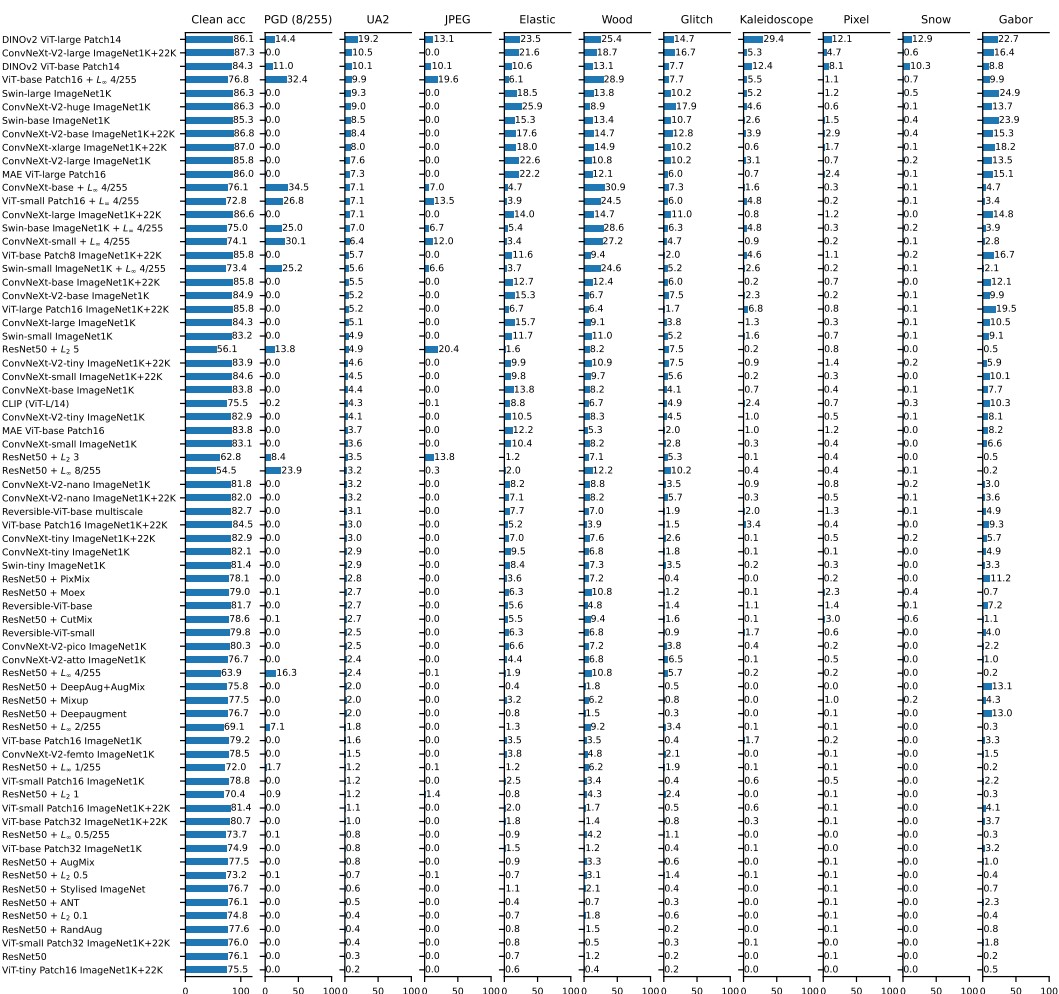

Figure 8: ImageNet UA2 performance under high distortion

## F.2    CIFAR-10

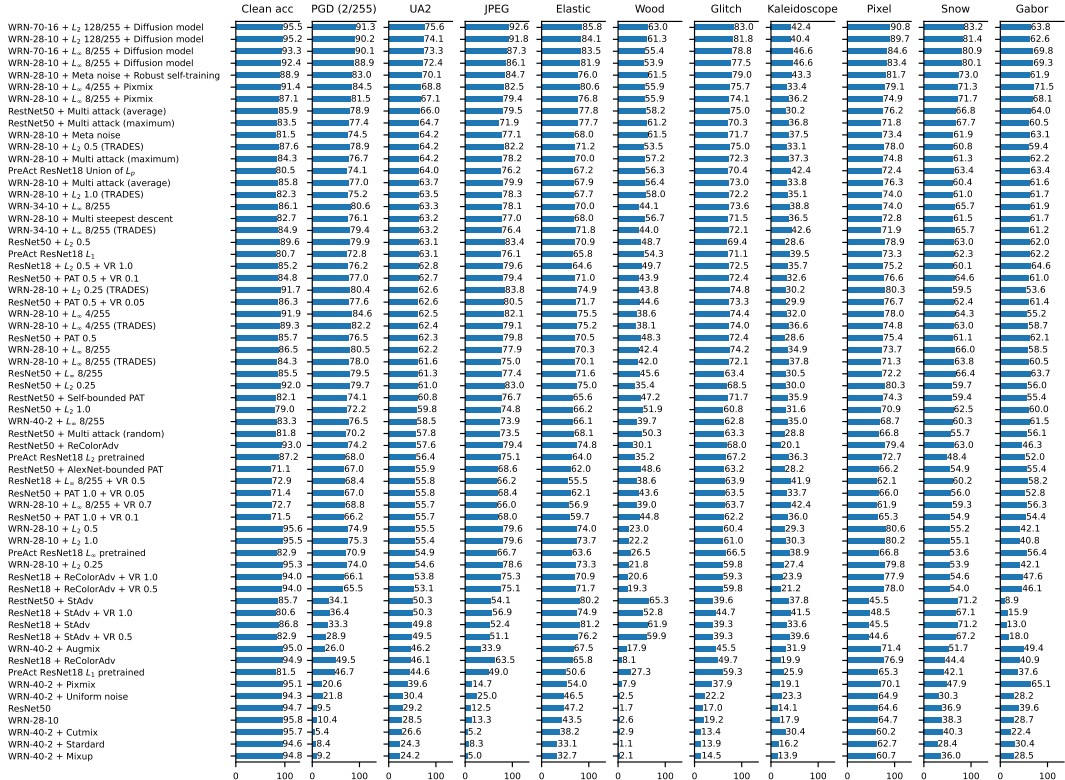

Figure 9: CIFAR-10 UA2 performance under low distortion.

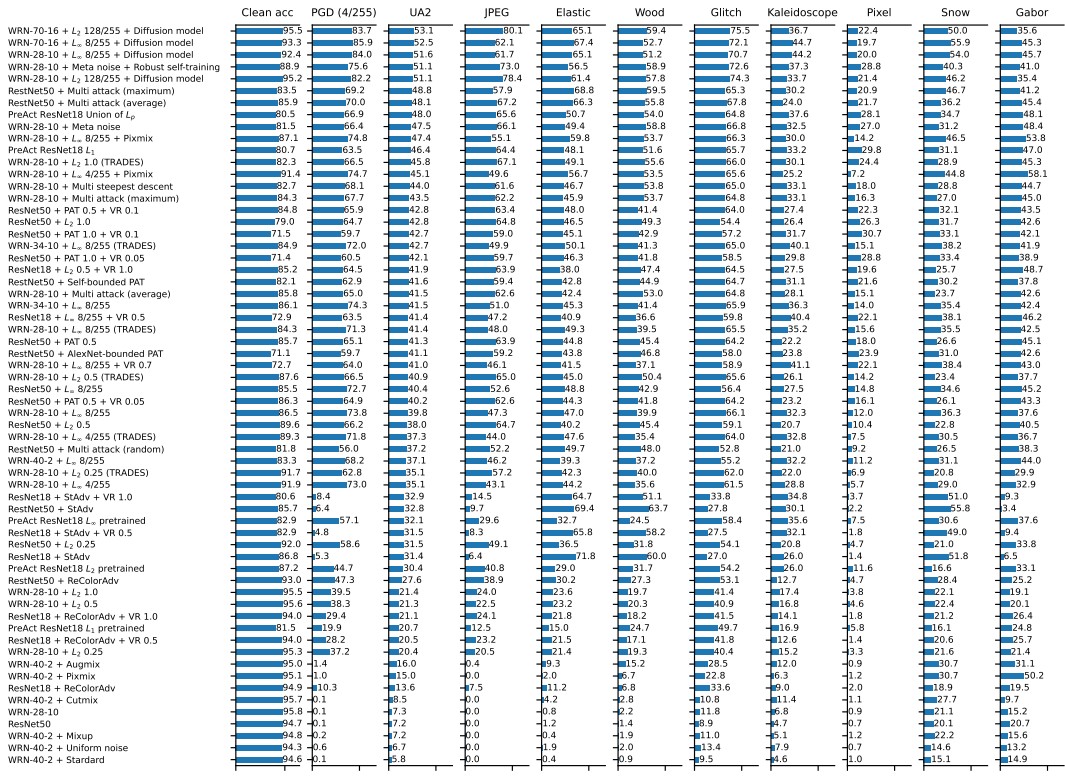

Figure 10: CIFAR-10 UA2 performance under medium distortion

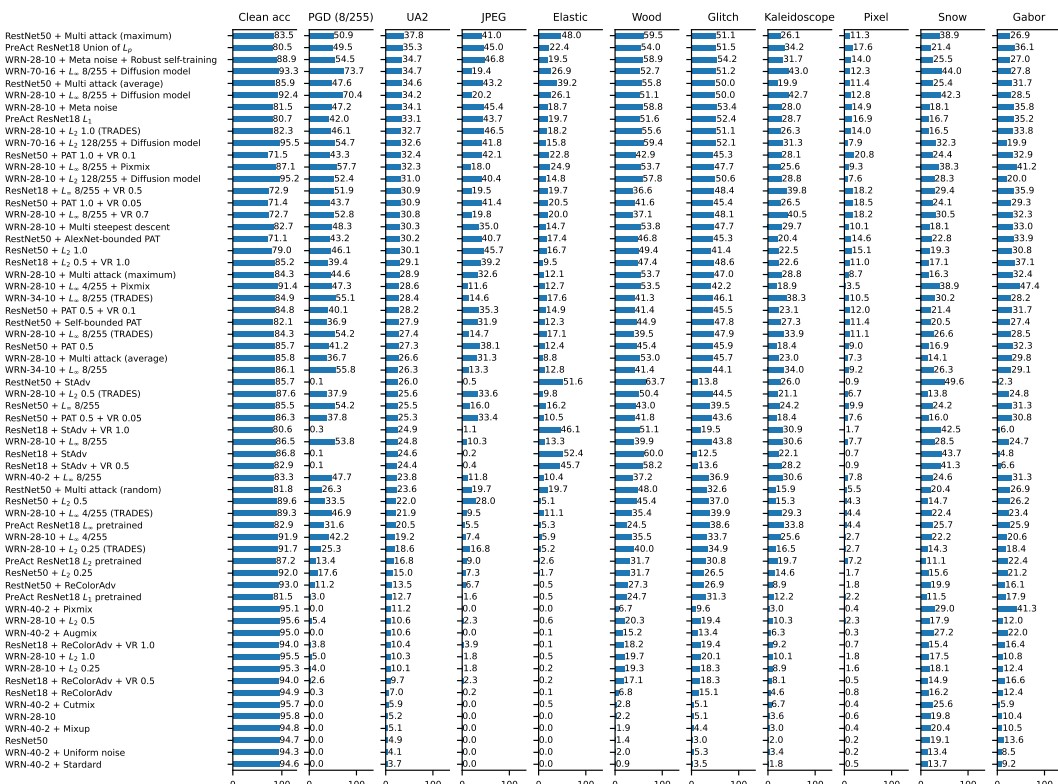

Figure 11: CIFAR-10 UA2 performance under high distortion

### F.3 IMAGENET100

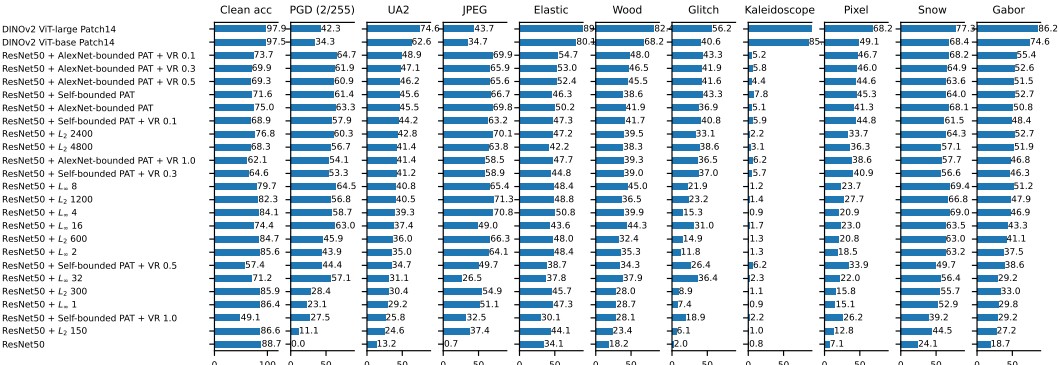

Figure 12: ImageNet100 UA2 performance under low distortion.

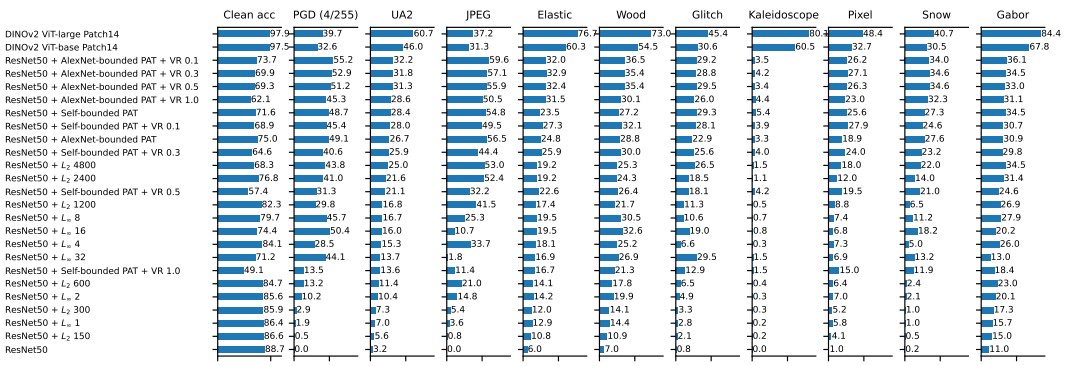

Figure 13: ImageNet100 UA2 performance under medium distortion

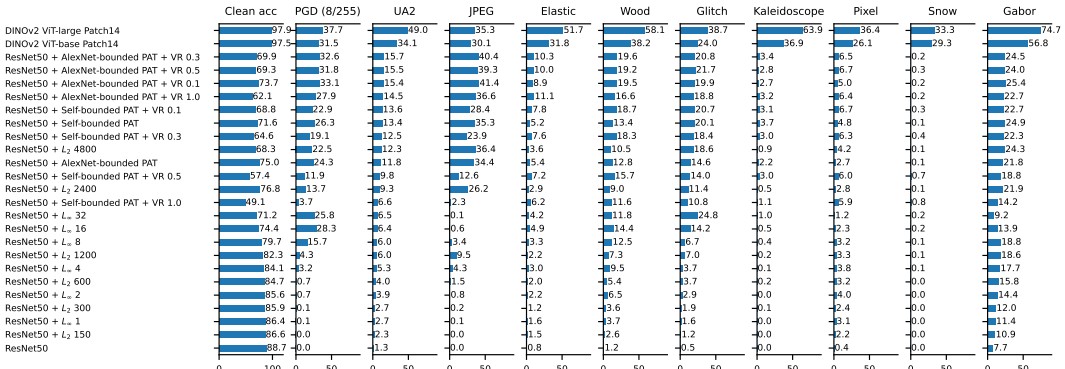

Figure 14: ImageNet100 UA2 performance under high distortion

## F.4 EXPLORING THE ROBUSTNESS OF DINOV2

Given the strong adversarial robustness of DINOv2 models under the PGD attack (Appendix F), we further evaluate the DINOv2 model under AutoAttack Croce & Hein (2020). Table 10 and Table 11 show that although for the robust ResNet50 model AutoAttack performs similarly to PGD, it is able to reduce the accuracy of DINOv2 models to 0.0% across all the distortion levels. Future work may benefite from applying the AutoAttack benchmark as a comparison point, instead of the base PGD adversary.

Table 10: Attacked accuracies of models on ImageNet

|  | ResNet50 + $L_\infty$ 8/255 | DINOv2 ViT-base Patch14 | DINOv2 ViT-large Patch14 |
|---|---|---|---|
| PGD (2/255) | 46.8% | 12.0% | 16.7% |
| APGD-CE (2/255) | 46.2% | 1.0% | 1.0% |
| APGD-CE + APGD-T (2/255) | 43.6% | 0.0% | 0.0% |
| PGD (4/255) | 38.9% | 11.4% | 15.3% |
| APGD-CE (4/255) | 37.9% | 0.9% | 0.8% |
| APGD-CE + APGD-T (4/255) | 33.8% | 0.0% | 0.0% |
| PGD (8/255) | 23.9% | 11.0% | 14.4% |
| APGD-CE (8/255) | 22.6% | 0.6% | 0.7% |
| APGD-CE + APGD-T (8/255) | 18.4% | 0.0% | 0.0% |

Table 11: Attacked accuracies of models on ImageNet100

|  | ResNet50 + $L_\infty$ 8/255 | DINOv2 ViT-base Patch14 | DINOv2 ViT-large Patch14 |
|---|---|---|---|
| PGD (2/255) | 64.5% | 34.3% | 42.3% |
| APGD-CE (2/255) | 64.4% | 17.6% | 20.0% |
| APGD-CE + APGD-T (2/255) | 64.1% | 0.0% | 0.0% |
| PGD (4/255) | 45.7% | 32.6% | 39.7% |
| APGD-CE (4/255) | 45.2% | 16.4% | 17.3% |
| APGD-CE + APGD-T (4/255) | 44.6% | 0.0% | 0.0% |
| PGD (8/255) | 15.7% | 31.5% | 37.7% |
| APGD-CE (8/255) | 14.7% | 15.5% | 14.5% |
| APGD-CE + APGD-T (8/255) | 13.6% | 0.0% | 0.0% |

## F.5 PERFORMANCE VARIANCE

As described in Section 4.1, we perform adversarial attacks by optimizing latent variables which are randomly initialized in our current implementation, so the adversarial attack's performance can be affected by the random seed for the initialization. To study the effect of random initializations, we compute the UA2 performances of three samples of two ImageNet models, ResNet50 and ResNet50 + $L_2$ 5. We observe the standard deviations of UA2 of these two models across 5 different seeds to be respectively 0.1% and 0.04% concluding that the variation in performance across the ImageNet dataset is minor.

## G IMAGES OF ALL ATTACKS ACROSS DISTORTION LEVELS

We provide images of all 19 attacks within the benchmark, across the three distortion levels.

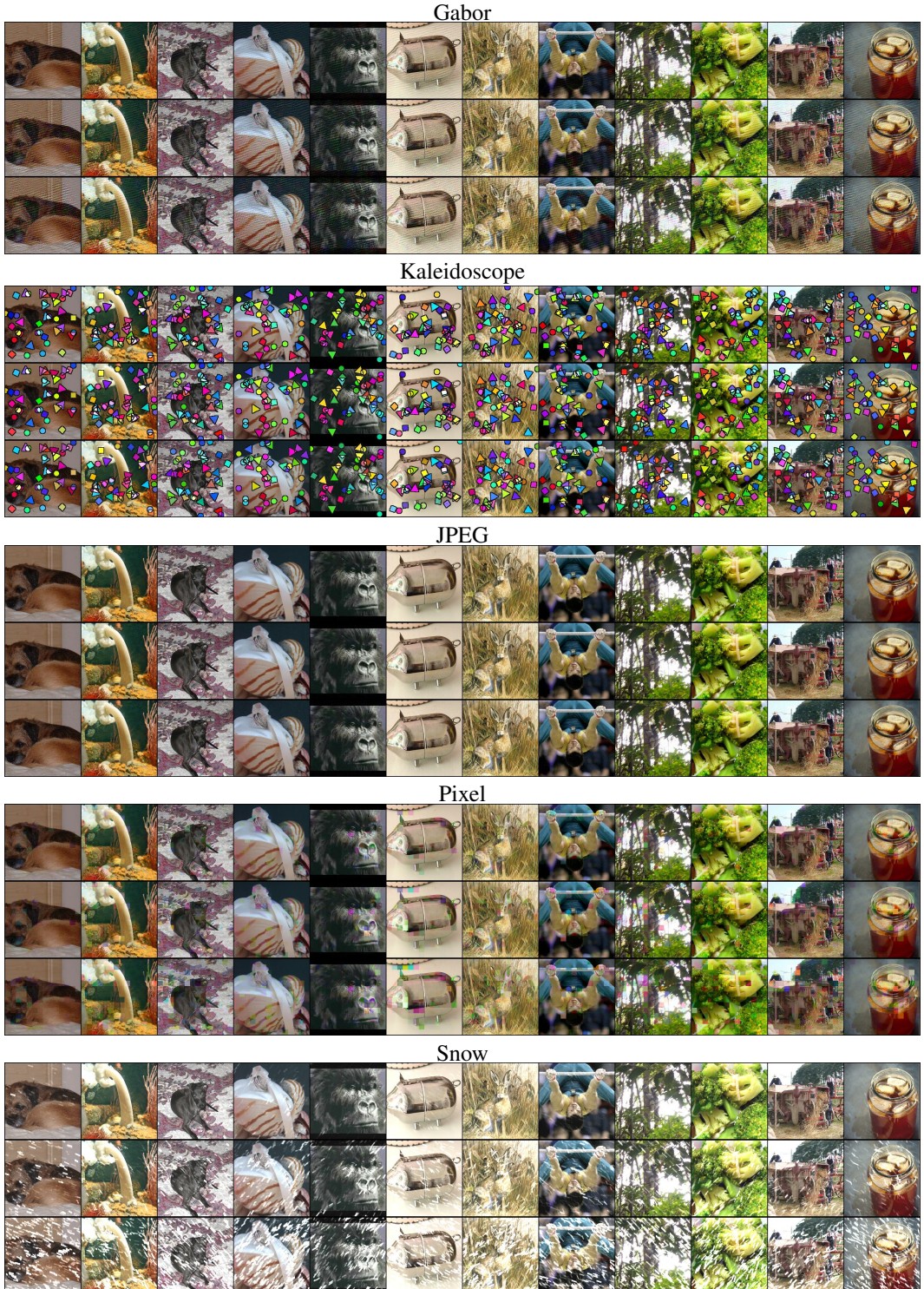

Figure 15: Attacked samples of low distortion (1st row), medium distortion (2nd row), and high distortion (last row) on a standard ResNet50 model

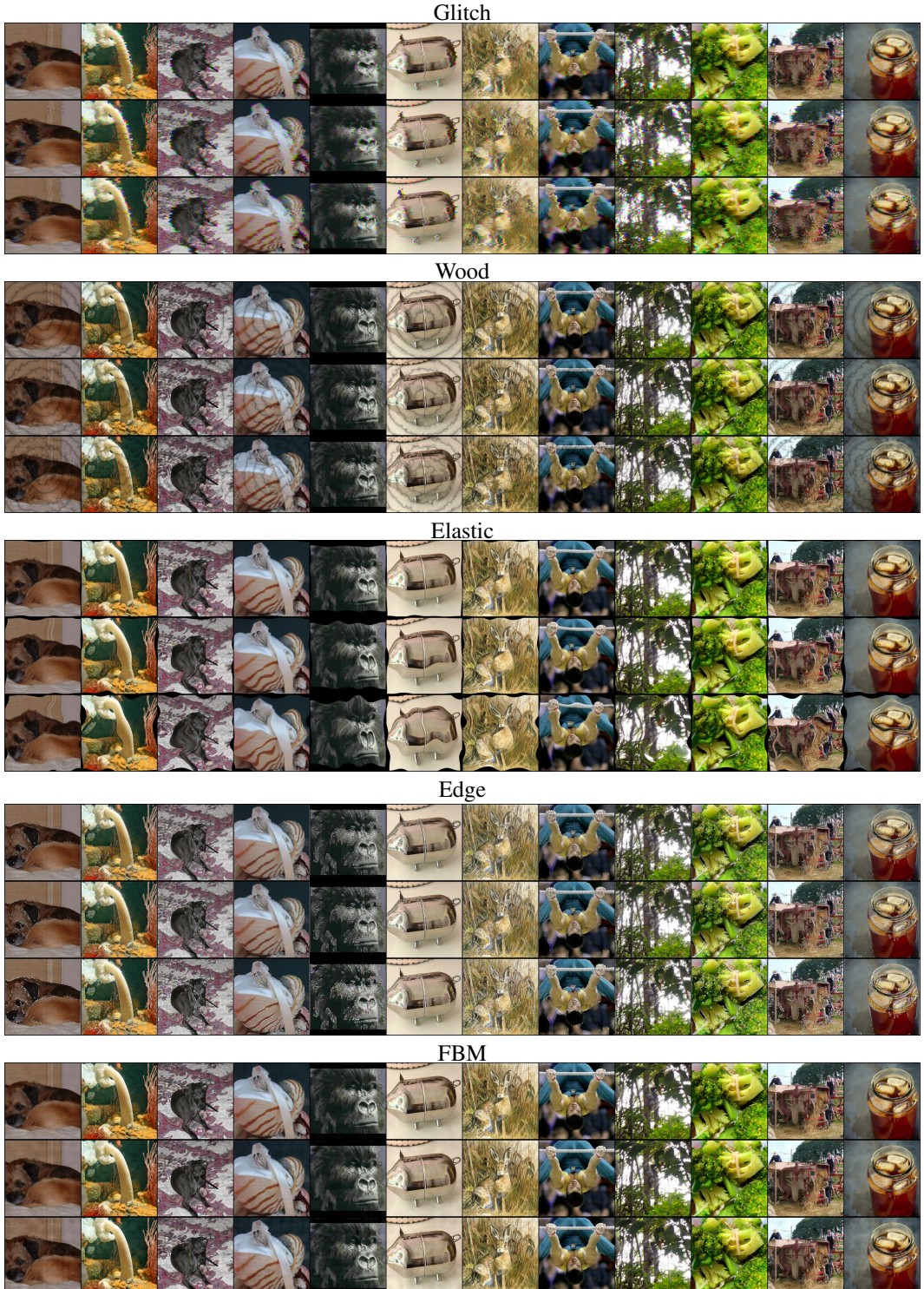

Figure 16: Attacked samples of low distortion (1st row), medium distortion (2nd row), and high distortion (last row) on a standard ResNet50 model

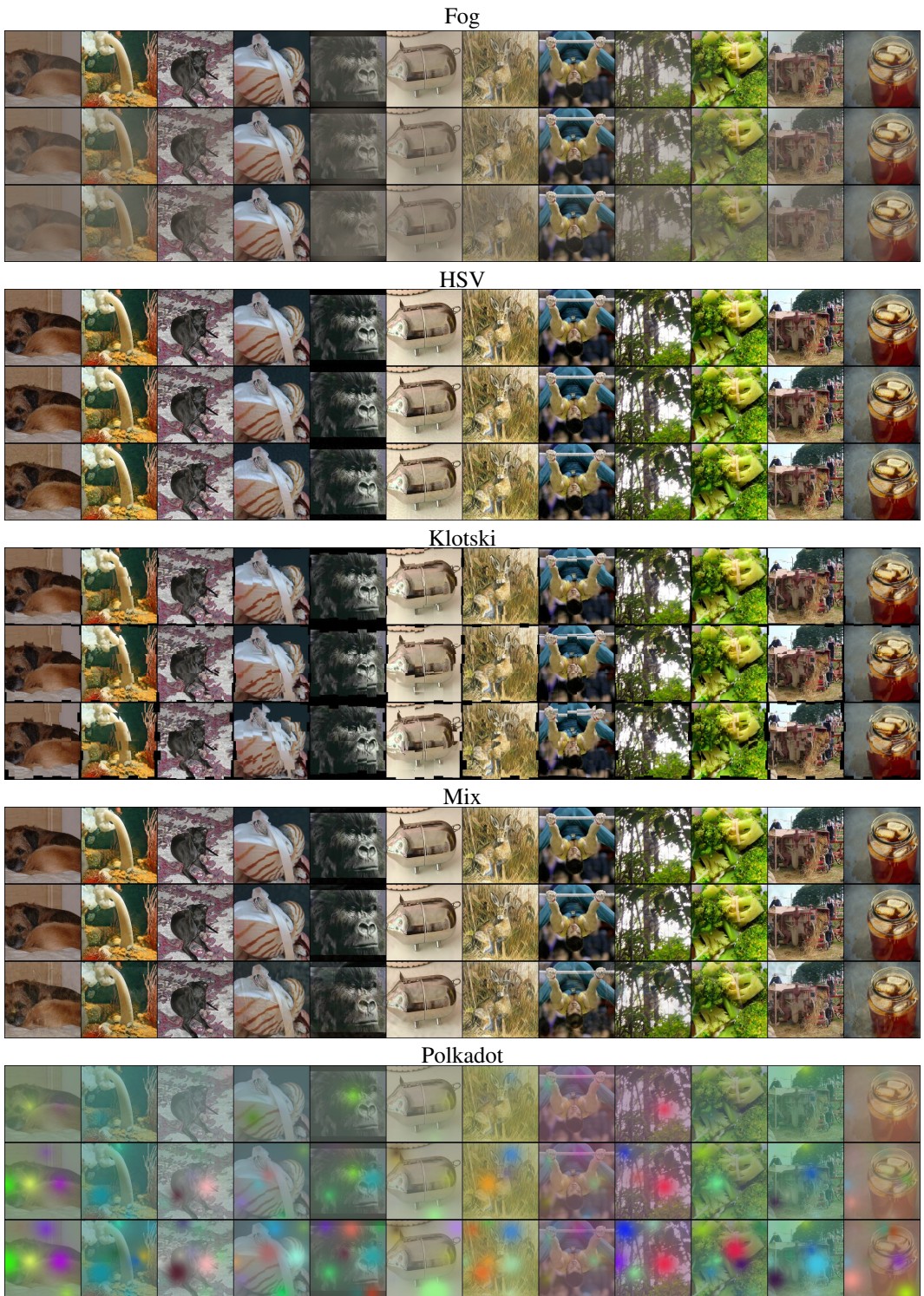

Figure 17: Attacked samples of low distortion (1st row), medium distortion (2nd row), and high distortion (last row) on a standard ResNet50 model

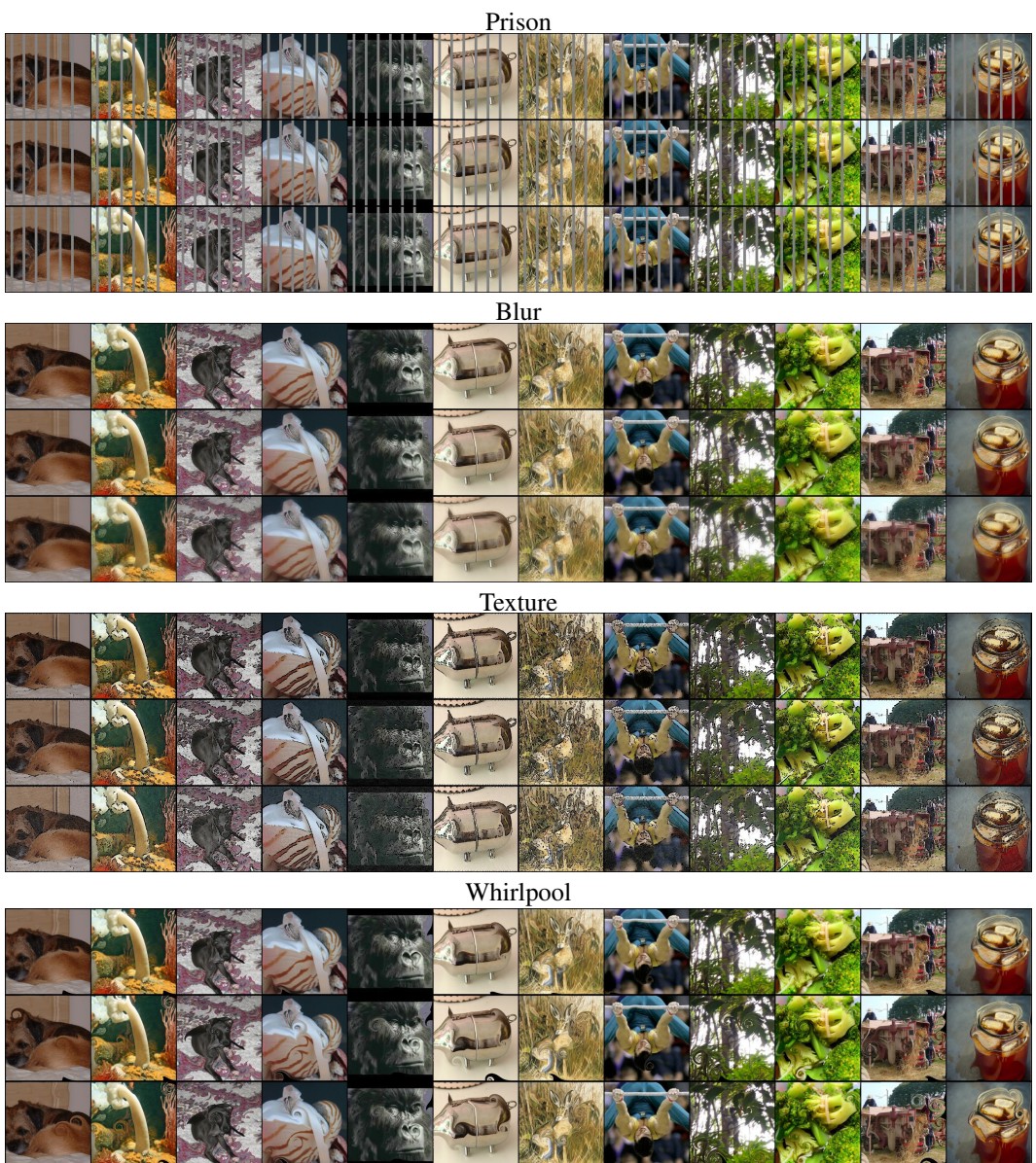

Figure 18: Attacked samples of low distortion (1st row), medium distortion (2nd row), and high distortion (last row) on a standard ResNet50 model

## H  SCALING BEHAVIOUR OF OUR ATTACKS

To see how our attacks perform across model scale, we make use of the ConvNeXt-V2 model suite (Woo et al., 2023) to test the performance of our attacks as we scale model size. We find that capacity improves performance across the board, but find diminishing returns to simply scaling up the architectures, pointing towards techniques described in Section 5.2.

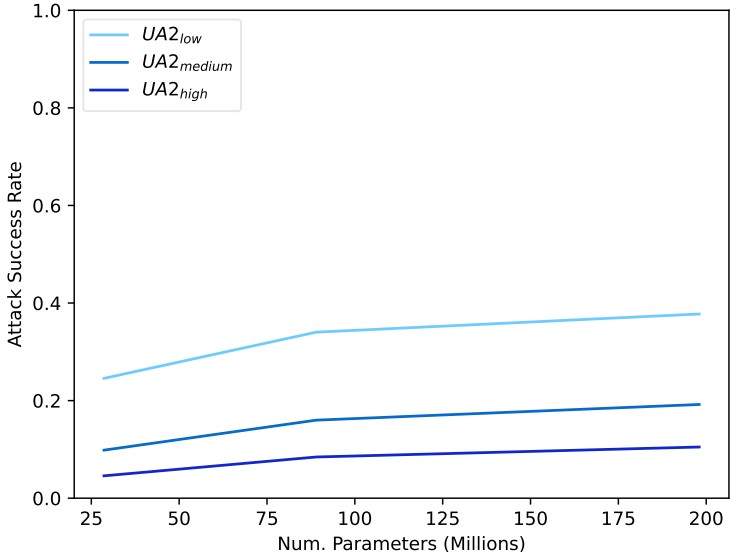

Figure 19: **Unforeseen Robustness across model scale.** We measure UA2 across model scale by evaluating the performance of ConvNeXt-V2 (Woo et al., 2023) models on ImageNet-UA, finding that scale improves performance, although the benchmark still provides a challenge to the largest models.

## I  IMAGENET-C AND UNFORESEEN ROBUSTNESS

Table 12: **Common corruptions and UA2** We compare performance on the ImageNet-C benchmark (mCE) to performance against both non-optimized and optimized versions of our attacks. We find that performance on the average-case robustness of ImageNet-C is correlated with performance on optimised attacks, while applying optimised versions favours the adversarially trained models.

| Model | UA2 (non-optimized) ↑ | mCE ↓ | UA2 ↑ |
|---|---|---|---|
| Resnet 50 | 55.2 | 76.7 | 1.6 |
| Resnet50 + AugMix | 59.1 | 65.7 | 3.5 |
| Resnet50 + DeepAug | 60.2 | **61.1** | 3.0 |
| Resnet50 + Mixup | 59.9 | 69.2 | 4.8 |
| Resnet50 + $L_2$, $(\varepsilon = 5)$ | 43.2 | 89.0 | 13.9 |
| Resnet50 + $L_\infty$, $(\varepsilon = 8/255)$ | 40.6 | 85.1 | 10 |

=

Table 13: **Distribution-shift benchmarks and UA2** Comparing performance on ImageNet-Sketch and ImageNet-R to performance against both non-optimized and optimized versions of UA2. We observe that performance on standard distribution shift benchmarks is correlated with performance on non-optimized UA2, while optimized UA2 settings favor models which have been trained for worst-case settings.

| Model | UA2 (non-optimised) | ImageNet-Sketch Acc. | ImageNet-R Acc. | UA2 |
|---|---|---|---|---|
| Resnet 50 | 55.2 | 24.1 | 36.2 | 1.6 |
| Resnet50 + AugMix | 59.1 | 28.5 | 41.0 | 3.5 |
| Resnet50 + DeepAug | **60.2** | **29.5** | **42.2** | 3.0 |
| Resnet50 + Mixup | 59.9 | 26.9 | 39.6 | 4.8 |
| Resnet50 + $L_2$, ($\varepsilon = 5$) | 43.2 | 24.2 | 38.9 | **13.9** |
| Resnet50 + $L_\infty$, ($\varepsilon = 8/255$) | 40.6 | 18.6 | 34.8 | 10 |

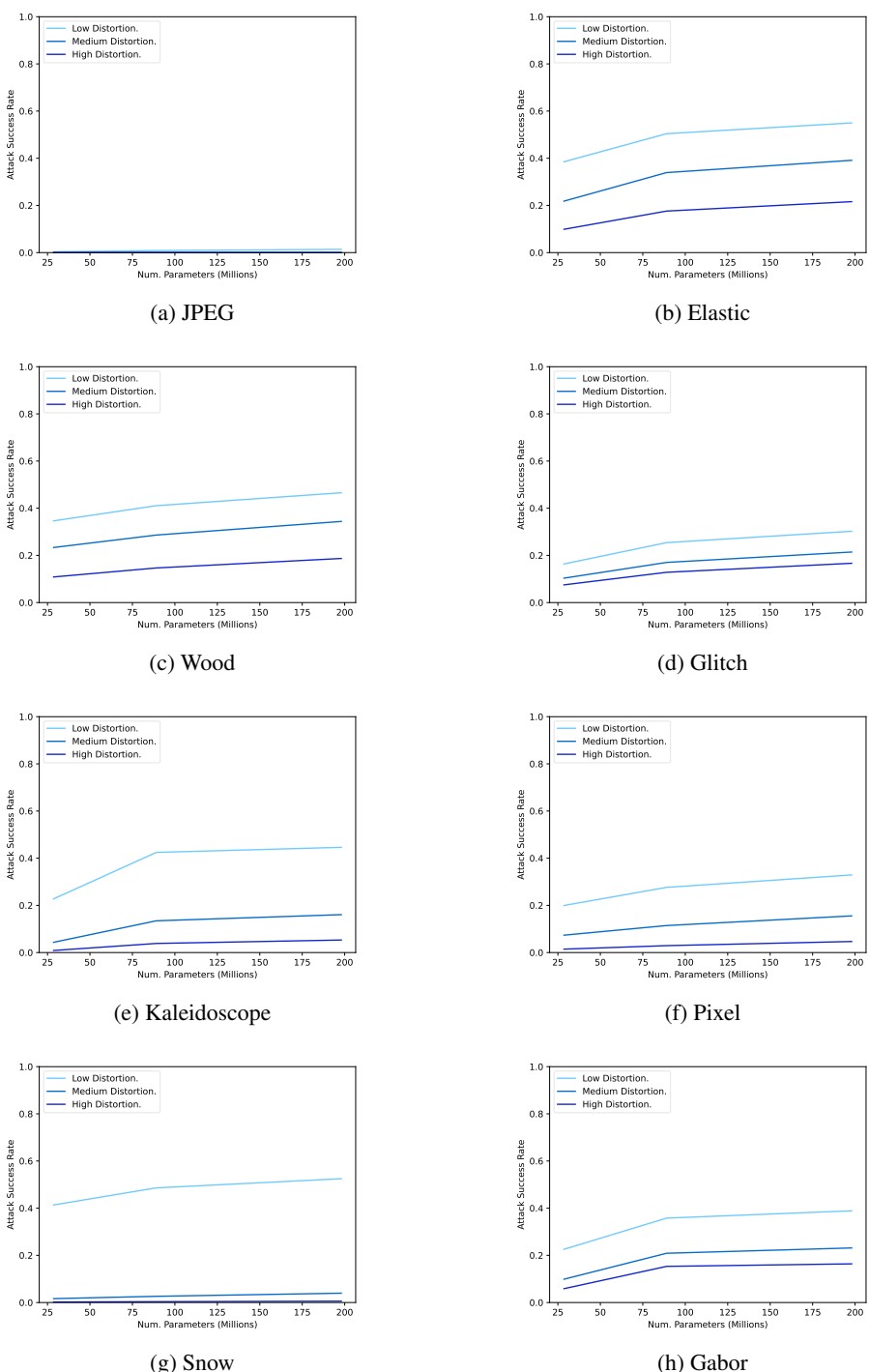

Figure 20: **Behaviour of core attacks across model scale.** We see the performance of the eight core attacks across the ConvNeXt-V2 model suite, with performance on attacks improving with model scale.

## J  BENCHMARKING NON- $L_p$ ADVERSARIAL TRAINING STRATEGIES

We wish to compare training strategies which have been specifically developed for robustness against both a variety of and unforeseen adversaries. To this end, we use Meta Noise Generation (Madaan et al., 2021b) as a strong multi-attack robustness baseline, finding that on CIFAR-10-UA this leads to large increases in robustness (Table 14). We also evaluate Perceptual Adversarial Training (Laidlaw et al., 2020) and Variational Regularization (Dai et al., 2022), two techniques specifically designed to achieve unforeseen robustness. We also evaluate combining PixMix and $L_p$ adversarial training. All of these baselines beat $L_p$ training.

Table 14: **Comparing alternative training strategies to $L_p$ baselines** We demonstrate that models trained using Meta Noise Generation (MNG) (Madaan et al., 2021b) improve over $L_p$ training baselines on CIFAR-10-UA.

| Training | Clean Acc. | UA2 |
|---|---|---|
| Standard | **95.8** | 7.4 |
| $L_\infty, \varepsilon = 8/255$ | 86.5 | 39.8 |
| $L_2, \varepsilon = 2$ | 95.5 | 21.4 |
| MNG | 88.9 | **51.1** |

**Meta Noise Generation (MNG) out-performs $L_p$ baselines.** We find that MNG, a technique original developed for multi-attack robustness shows a 11.3% increase in UA2 on CIFAR-10-UA, and PAT shows a 3.5% increase in UA2.

Table 15: **Specialised Unforseen robustness training strategies.** We see that ImageNet-UA PAT (Laidlaw et al., 2020) and PAT-VR (Dai et al., 2022)trained ResNet50s improve over $L_p$ baselines. Selected $L_p$ models are the best Resnet50s from the bench-marking done in Figure 7, and for computational budget reasons they are trained on a 100-image subset of ImageNet, constructed by taking every 10th class.

| Training | Clean Acc. | UA2 |
|---|---|---|
| Standard | **88.7** | 3.2 |
| $L_\infty, \varepsilon = 8/255$ | 79.7 | 17.5 |
| $L_2, \varepsilon = 4800/255$ | 71.6 | 25.0 |
| PAT | 75.0 | 26.2 |
| PAT-VR | 69.4 | **29.5** |

Table 16: **PixMix and $L_p$ training.** We compare UA2 performance on CIFAR-10 of models trained with PixMix and adversarial training. Combining PixMix with adversarial training results in large improvements in UA2, demonstrating an exciting future direction for improving unforeseen robustness. All numbers denote percentages, and $L_\infty$ training was performed with the TRADES algorithm.

| Model | Clean Acc. | UA2 |
|---|---|---|
| WRN-40-2 + PixMix | **95.1** | 15.00 |
| WRN-28-10 + $L_\infty$ 4/255 | 89.3 | 37.3 |
| WRN-28-10 + $L_\infty$ 4/255 + PixMix | 91.4 | **45.1** |
| WRN-28-10 + $L_\infty$ 8/255 | 84.3 | 41.4 |
| WRN-28-10 + $L_\infty$ 8/255 + PixMix | 87.1 | **47.4** |

## K  HUMAN STUDY OF SEMANTIC PRESERVATION

Table 17: **Results of user study.** We run a user study on the 200 class subset of ImageNet presented as part of ImageNet-R (Hendrycks et al., 2021), assessing the multiple-choice classification accuracy of human raters, allowing raters to choose certain images as corrupted. We use 4 raters per label and take a majority vote, finding high classification accuracy across all attacks.

| Attack Name | Correct | Corrupted or Ambiguous |
|---|---|---|
| Clean | 95.4 | 4.2 |
| Elastic | 92.0 | 2.0 |
| Gabor | 93.4 | 4.0 |
| Glitch | 80.2 | 16.0 |
| JPEG | 93.4 | 0.6 |
| Kaleidescope | 93.0 | 6.2 |
| Pixel | 92.6 | 1.8 |
| Snow | 90.0 | 3.2 |
| Wood | 91.4 | 1.8 |
| Adversarial images average | 91.2 | 4.5 |

We ran user studies to compare the difficulties of labeling the adversarial examples compared to the clean examples. We observe that under our distribution of adversaries users experience a 4.2% drop in the ability to classify. This highlights how overall humans are still able to classify over 90% of the images, implying that the attacks have not lost the semantic information, and hence that models still have room to grow before they match human-level performance on our benchmark.

In line with ethical review considerations, we include the following information about our human study:

- **How were participants recruited?** We made use of the surgehq.ai platform to recruit all participants.
- **How were the participants compensated?** Participants were paid at a rate of $0.05 per label, with an average rating time of 4 seconds per image—ending at an average rate of roughly $45 hour.
- **Were participants given the ability to opt out?** All submissions were voluntary.
- **Were participants told of the purpose of their work?** Participants were told that their work was being used to "validate machine learning model performance".
- **Was any data or personal information collected from the participants?** No personal data was collected from the participants.
- **Was there any potential risks done to the participants?** Although some ImageNet classes are sometimes known to contain elicit or unwelcome content Prabhu (2019). Our 100-class subset of ImageNet purposefully excludes such classes, and as such participants were not subject to any undue risks or personal harms.

## Adversarial Images Classification

This work is used to validate machine learning model performance and your participation is voluntary. You're free to stop the task at any point in time. You'll be shown an image. One of the labels is indeed present in the image please select the correct one. If you're unfamiliar with a label take a second to search for it on google images. Please let us know if this happens often.

The image may however be too corrupted in which case select that it is too corrupted. **Please avoid using corrupted label unless necessary.**

Thanks!

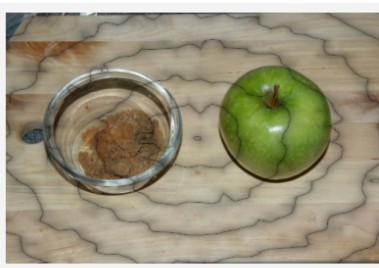

Select which label is present in the image or if the image is too corrupted.

○ Granny Smith (type of eating apple)

○ pretzel (type of pretzel)

○ pufferfish (type of fish)

○ saxophone (type of musical instrument)

○ accordion (type of musical instrument)

○ Image is too corrupted

Next preview

Figure 21: **Interface of participants.** We demonstrate the interface which was provided to the participants of the study, involving the selection of correct classes from our 100-class subset of ImageNet.

```
This work is used to validate machine learning model performance and your
participation is voluntary.  You're free to stop the task at any point in
time.
You'll be shown an image.  One of the labels is indeed present in the
image please select the correct one.  If you're unfamiliar with a label
take a second to search for it on google images.  Please let us know if
this happens often.
The image may however be too corrupted in which case select that it is
too corrupted.  Please avoid using corrupted label unless necessary.
Thanks!
```

Figure 22: **Instructions given to the participants.** Above is a list of the instructions which were given to the participants in the human study.

## L    CORRELATION OF $L_p$ ROBUSTNESS AND ImageNet-UA

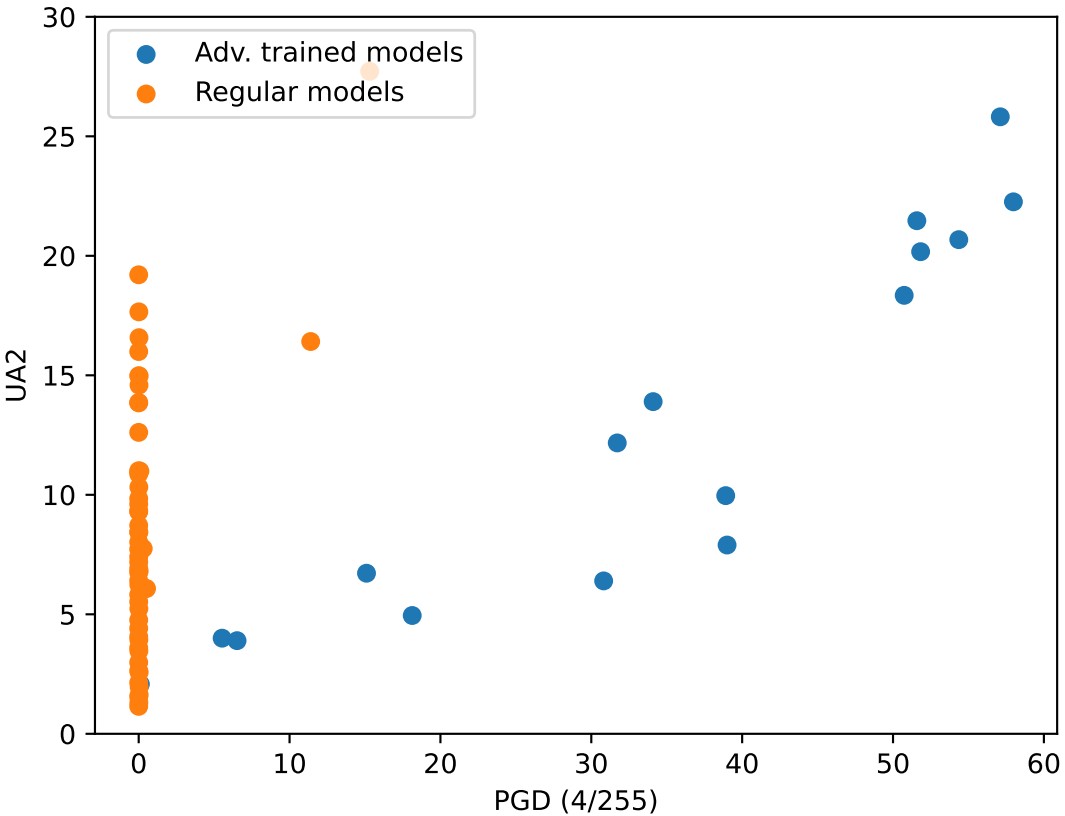

Figure 23: $L_p$ **robustness correlates with UA2.** Across our benchmark, for adversarially trained models $L_p$ robustness correlates with UA2 - however, several models trained without adversarial training still improve on UA2.

## M    GRID SEARCH VS. GRADIENT-BASED SEARCH

Table 18: **Comparing gradient-based search to grid-based search** We compare the performance of optimising with a randomised grid-based search using 1000 forward-passes per datapoint, finding that our gradient-based methods perform a lot better than this compute-intensive baseline.

| Optimisation Technique | UA2 |
|---|---|
| Randomized grid search | 74.1 |
| Gradient-based search (ours) | 7.2 |

# N    TRANSFER ATTACKS

Table 19 shows the transfer-attack performances across various source and target models based on 1000 test samples. We observe that while the transfer attacks are not as effective as white-box attacks, they consistently outperform baseline unoptimized attacks where the perturbations are randomly initialized (Table 20).

Table 19: Transfer attack performance

| | Clean Acc. | PGD | UA2 | JPEG | Elastic | Wood | Glitch | Kal. | Pixel | Snow | Gabor |
|---|---|---|---|---|---|---|---|---|---|---|---|
| ResNet50 (**source model**) | 75.2 | 0 | 13.2 | 0 | 22.2 | 30.8 | 10 | 4.3 | 4.8 | 3.1 | 30.4 |
| ViT-small Patch16 ImageNet1K | 78.5 | 73.1 | 59.99 | 75 | 62.7 | 69.9 | 46 | 48 | 62.8 | 55.5 | 60 |
| ConvNeXt-V2-tiny ImageNet1K | 82.1 | 74.8 | 67.66 | 77.1 | 69 | 75.9 | 54 | 60 | 73.6 | 65.2 | 66.5 |
| Swin-small ImageNet1K $+L_\infty$ 4/255 | 71.1 | 70.6 | 50.39 | 70.9 | 56.7 | 65.8 | 34.8 | 10.7 | 59.3 | 48.4 | 56.5 |
| ResNet50 | 75.2 | 67.9 | 43.19 | 70.1 | 53.1 | 57.7 | 30.1 | 5.4 | 53.3 | 38.1 | 37.7 |
| ViT-small Patch16 ImageNet1K (**source model**) | 78.5 | 0 | 6.51 | 0 | 8.2 | 12.7 | 0.5 | 4.7 | 2.1 | 0.8 | 23.1 |
| ConvNeXt-V2-tiny ImageNet1K | 82.1 | 75.7 | 67.3 | 78.6 | 68.5 | 72.8 | 56.4 | 59.9 | 70.1 | 65.1 | 67 |
| Swin-small ImageNet1K $+ L_\infty$ 4/255 | 71.1 | 70.5 | 50.11 | 70.9 | 57.1 | 65.1 | 35 | 10.8 | 59.5 | 48 | 54.5 |
| ResNet50 | 75.2 | 67.8 | 42.06 | 68.3 | 51 | 55.7 | 31.7 | 5.8 | 51.7 | 32.1 | 40.2 |
| ViT-small Patch16 ImageNet1K | 78.5 | 74.7 | 57.31 | 75 | 60 | 69 | 42 | 46.8 | 57.2 | 50.2 | 58.3 |
| ConvNeXt-V2-tiny ImageNet1K (**source model**) | 82.1 | 0 | 12.15 | 0 | 23.2 | 22.3 | 7.4 | 3.5 | 6 | 0.6 | 34.2 |
| Swin-small ImageNet1K $+ L_\infty$ 4/255 | 71.1 | 71.2 | 50.1 | 71.2 | 56.1 | 65 | 37.8 | 10.7 | 59.1 | 45 | 55.9 |
| ResNet50 | 75.2 | 64 | 36.95 | 61.8 | 42.5 | 57.8 | 15.6 | 5.4 | 45.3 | 29.2 | 38 |
| ViT-small Patch16 ImageNet1K | 78.5 | 66.9 | 53.3 | 70.6 | 51.4 | 68.2 | 23.8 | 47.1 | 58.4 | 44.2 | 62.7 |
| ConvNeXt-V2-tiny ImageNet1K | 82.1 | 75.5 | 65.26 | 75.7 | 64.7 | 74.5 | 46.1 | 58.2 | 72.3 | 63.6 | 67 |
| Swin-small ImageNet1K $+ L_\infty$ 4/255 (**source model**) | 71.1 | 53.8 | 21.4 | 42 | 17.9 | 42.3 | 5.1 | 5.1 | 7.6 | 3.4 | 47.8 |

Table 20: Unoptimized attack performance

| | Clean Acc. | PGD | UA2 | JPEG | Elastic | Wood | Glitch | Kal. | Pixel | Snow | Gabor |
|---|---|---|---|---|---|---|---|---|---|---|---|
| ResNet50 | 75.2 | 74.1 | 56.44 | 74.3 | 62.8 | 55.7 | 55.8 | 6.3 | 74.1 | 74.8 | 47.7 |
| ViT-small Patch16 ImageNet1K | 78.5 | 78 | 69.19 | 78 | 70.2 | 70.2 | 65.4 | 47.7 | 77.3 | 78.6 | 66.1 |
| ConvNeXt-V2-tiny ImageNet1K | 82.1 | 82.2 | 74.74 | 82.2 | 75.2 | 74.4 | 69.7 | 60.7 | 81.5 | 81.4 | 72.8 |
| Swin-small ImageNet1K $+ L_\infty$ 4/255 | 71.1 | 71.3 | 58.19 | 71.6 | 62 | 63.4 | 58 | 10.2 | 70.9 | 71.7 | 57.7 |

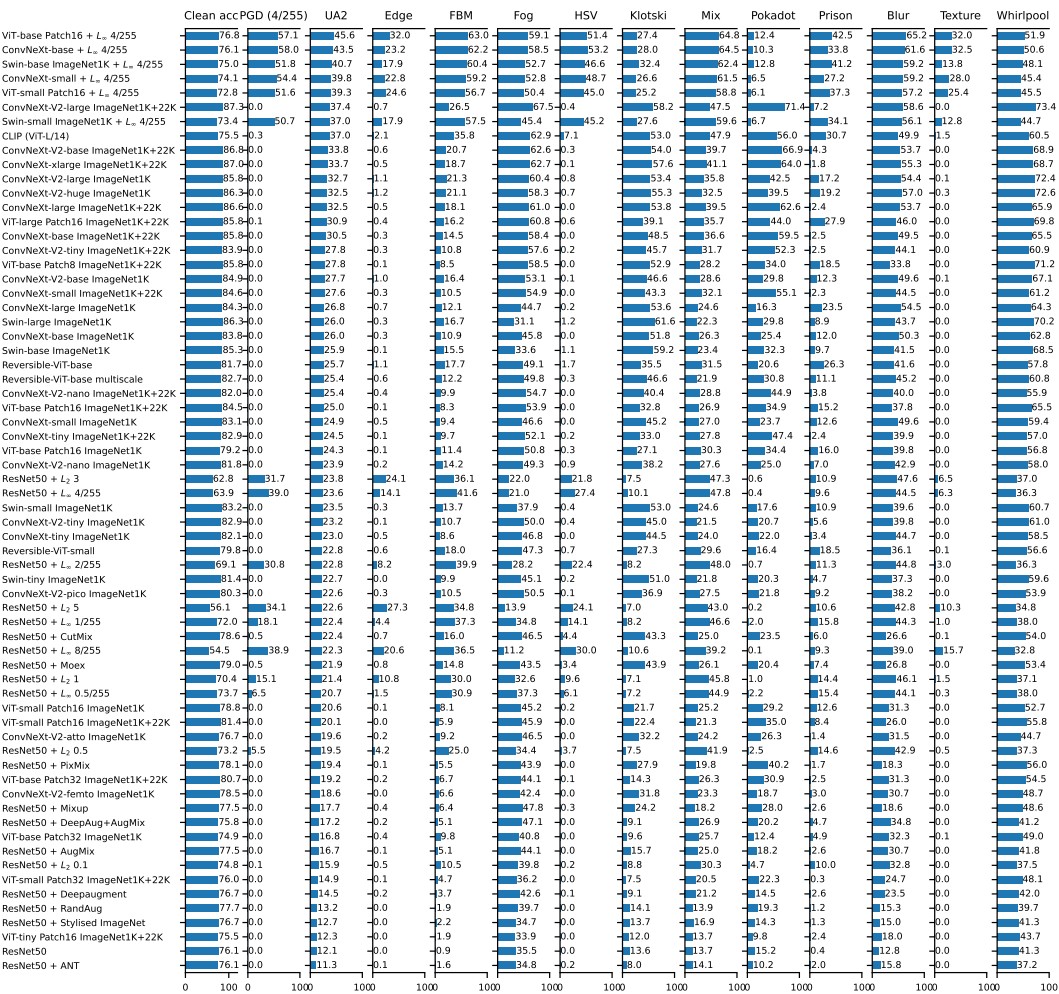

Figure 24: ImageNet UA2 performance under extra attacks in medium distortion

