# OpenReview forum: "Evaluating Robustness to Unforeseen Adversarial Attacks"
_ICLR.cc/2024/Conference — Submitted to ICLR 2024_

### Official Review · Reviewer_hiUg · 2023-10-13

**Soundness:** 4 excellent
**Presentation:** 2 fair
**Contribution:** 3 good
**Rating:** 8
**Confidence:** 4

**Summary:**

This paper introduces a set of 18 new attacks for evaluating unforeseen robustness and propose a metric UAR2 which uses these attacks for measuring the performance of existing models.  The authors then evaluate existing models using this UAR2 metric to determine the impact of adversarial training and other defenses for the task of unforeseen robustness, impact of data augmentation, and impact pretraining and regularization.

**Strengths:**

I think that the direction of unforeseen robustness is an interesting and important problem in adversarial robustness which currently has not gained much attention.  I think that the proposed benchmark is a useful tool for evaluating current models, understanding how different training choices impact unforeseen robustness, and can motivate more research in the field.  I also find the presentation to be clear overall and am impressed by the large scope of evaluations presented across the paper and Appendix.

**Weaknesses:**

- I think that the paper can reference the work [1] which also looks at the problem of benchmarking unforeseen robustness and performs several ablations on the impact of factors like using synthetic data during adversarial training on unforeseen robustness.  This work differs from the MultiRobustBench paper as the MultiRobustBench paper focuses mainly on robustness for CIFAR-10 due to computational constraints and does not propose new attacks.
- There are some results that are referenced in the main paper (mainly those which compare existing defenses against unforeseen attacks and the MNG approach) which are in the Appendix.  If possible, it would be good to move them into the main paper since I think these evaluations of adversarial training approaches are quite important.
- It would also be interesting to also have a few more evaluations of existing defenses tailored to unions of lp attack robustness outside of the MNG approach (ie. the approach in [2] or [3] might be able to be used with ImageNet, or evaluating approaches in [4] or [5] with CIFAR-10) in order to have a better understanding of how well these approaches for multiattack robustness generalize to unforeseen attacks

[1] Dai, S., Mahloujifar, S., Xiang, C., Sehwag, V., Chen, P. &amp; Mittal, P.. (2023). MultiRobustBench: Benchmarking Robustness Against Multiple Attacks. **Proceedings of the 40th International Conference on Machine Learning**, in **Proceedings of Machine Learning Research** 202:6760-6785 Available from https://proceedings.mlr.press/v202/dai23c.html.
[2] Sriramanan, G., Gor, M., & Feizi, S. (2022). Toward Efficient Robust Training against Union of $\ell_p $ Threat Models. **Advances in Neural Information Processing Systems**, 35, 25870-25882.
[3] Croce, F., & Hein, M. (2022, June). Adversarial Robustness against Multiple and Single $ l_p $-Threat Models via Quick Fine-Tuning of Robust Classifiers. In International Conference on Machine Learning (pp. 4436-4454). PMLR.
[4] Maini, P., Wong, E., & Kolter, Z. (2020, November). Adversarial robustness against the union of multiple perturbation models. In International Conference on Machine Learning (pp. 6640-6650). PMLR.
[5] Tramer, Florian, and Dan Boneh. "Adversarial training and robustness for multiple perturbations." Advances in neural information processing systems 32 (2019).

**Questions:**

How different are the different forms of robustness tested across the benchmark?  For example, does robustness against something like glitch also give some robustness against JPEG attacks?

---

> ### Author Response · Authors · 2023-11-21
> **Author Response**
>
> Thank you for your careful analysis of our work. We hope the following response addresses your concerns.
>
> >I think that the paper can reference the work [1] which also looks at the problem of benchmarking unforeseen robustness and performs several ablations on the impact of factors like using synthetic data during adversarial training on unforeseen robustness. This work differs from the MultiRobustBench paper as the MultiRobustBench paper focuses mainly on robustness for CIFAR-10 due to computational constraints and does not propose new attacks.
>
> We have added this paper to the related work in the updated paper. Thank you for your suggestion.
>
>
> >There are some results that are referenced in the main paper (mainly those which compare existing defenses against unforeseen attacks and the MNG approach) which are in the Appendix. If possible, it would be good to move them into the main paper since I think these evaluations of adversarial training approaches are quite important.
>
> We agree that these results are important and will incorporate them in the main body for the camera-ready version. Thank you for your suggestion.
>
> >It would also be interesting to also have a few more evaluations of existing defenses tailored to unions of lp attack robustness outside of the MNG approach (ie. the approach in [2] or [3] might be able to be used with ImageNet, or evaluating approaches in [4] or [5] with CIFAR-10) in order to have a better understanding of how well these approaches for multiattack robustness generalize to unforeseen attacks
>
> These models are non-public, but we have reached out to the paper authors. Due to the short timelines of this rebuttal period they have yet to get back to us, but we plan to add an analysis of these additional methods once we have access to the models. (UPDATE: See below for results on Sriramanan et al. [2]).

---

> ### Author Response · Authors · 2023-11-22
>
> Update: We have updated the manuscript to include results of models from Sriramanan, G., Gor, M., & Feizi, S. (2022). "Toward Efficient Robust Training against Union of Lp Threat Models". These results have been added to Figures 9, 10, and 11 with entry names "PreAct ResNet18 L1" and "PreAct ResNet18 Union of Lp". They perform quite well, with "PreAct ResNet18 Union of Lp" obtaining the second-highest UA2 on CIFAR-10-UA under the "high" distortion. Thank you for your suggestion.

---

> > ### Comment · Reviewer_hiUg · 2023-11-23
> >
> > Thank you for the response, I believe all my points were addressed and have raised my score to an 8.

---

### Official Review · Reviewer_PXCR · 2023-10-29

**Soundness:** 3 good
**Presentation:** 3 good
**Contribution:** 2 fair
**Rating:** 3
**Confidence:** 4

**Summary:**

This paper proposes a benchmark to evaluate robustness against unforeseen adversarial attacks. To model unforeseen attacks, this benchmark includes 19 non-Lp adversarial example generation methods. These attacks are designed to be differentiable so the attacker can leverage gradient-based optimizers like PGD. Evaluations provide insights into several research questions, such as how Lp robustness measures relate to unforeseen robustness, what techniques can improve unforeseen robustness, and how classic progress on CV partially improved the unforeseen robustness.

**Strengths:**

### Originality

* The proposed benchmark and its attacks are mostly novel.
* The evaluation provides several interesting insights into the new notion of adversarial robustness.

### Quality

* Evaluations are sound.
* Methodology is sound, it is good to see that non-Lp attacks can optimize worst-case perturbation where the Lp constraint is on some internal parameters.

### Clarity

* The presentation is clear and easy to follow.

### Significance

* Studying robustness against unforeseen attacks is important and necessary, given that most focus is on Lp-norm robustness.

**Weaknesses:**

### Originality

N/A

### Quality

**Q1: Unclear role of the proposed benchmark in the defense's development.**

This benchmark studies the problem of "unforeseen" attacks by labeling a set of new attacks as "unforeseen," which has a nature limitation in the sense that once these attacks are evaluated, they are no longer "unseen." From the current presentation, this benchmark should be evaluated once and dropped right away -- the defense should not explicitly optimize toward these attacks. For example, if a defense optimizes on some of these attacks, would they obtain higher robustness against the other remaining attacks? Would this action be treated as cheating by training on the test set?

To make the benchmark more useful, I feel it important to develop a validation/test split from these attacks. That is, the progress should be made to optimize against one subset of attacks, and evaluate their robustness against another subset of attacks. Only in this case one can really claim that they are improving the "unforeseen robustness" with this benchmark. This paper includes some evaluation by adversarial training with these attacks, and the performance indeed improves. However, this is obvious and cannot provide any confidence to even newer attacks, as from this paper's own argument, those newer attacks are now unseen compared with this benchmark's now-seen attacks. In this case, if the validation/test concept were applied, I assume that the paper would discover (again) that adversarial training does not provide unforeseen robustness, even if the model was trained with these "unforeseen" attacks.

### Clarity

* Typos like nineteen vs eighteen, how many attacks indeed?

### Significance

**Q2: Does this benchmark really model the robustness against unforeseen attacks by including 19 attacks?**

The importance of studying robustness against unforeseen attacks is that test-time attacks may not be observed at the training time. While this benchmark includes a wide range of empirical attacks, it is unclear if the measured robustness against these 19 now-seen attacks would generalize to robustness against other unforeseen attacks outside these 19 attacks. Yet, this is the whole point that matters in understanding unforeseen robustness. I wonder if a defense would always be more robust to "unforeseen" attacks if they demonstrate higher robustness against these 19 attacks. On the other hand, for the purpose of demonstrating non-robustness, if a defense observes low robustness against attack A, would this imply that the defense is not robust against attack B?

**Q3: The attacks are manually selected, which limits the benchmark's impact.**

As far as I understand, these attacks are manually constructed without any underlying systematic concept. As a result, the benchmark's outcome only provides "some" sense of unforeseen robustness against a fixed set of "manually selected" attacks that are interpreted as an "unforeseen" test set. The measured robustness cannot provide a confident claim that the defense will also have true robustness against unforeseen attacks. Note that I am not criticizing these attacks (their differentiable nature made a good case for different worst-case perturbations), but this manual design limits the significance of this benchmark.

It is suggested to systematize or even automate the attack's choice so that they can suggest some generalization. For example, these attacks are generally perturbations along different semantic directions on the image manifold. Right now such directions are manually chosen, is it possible to systematize such directions (even if they do not have very semantic interpretations) so they would exhibit some generalization to other directions?

**Questions:**

I appreciate the good design of attacks and frameworks, but I am concerned about the significance of this work, as the benchmark cannot rigorously make a true claim for "robustness against unforeseen attacks" (Q2) due to the lack of validation/test split (Q1) or a systematic design that functions similarly (Q3).

### Post-Rebuttal Update

**The role of validation set.**

> Therefore, we don't use the validation attacks to tune hyperparameters or perform adversarial training. We provide the validation attacks primarily to help future research on this problem.

However, it is highlighted in the updated draft that *"We leave the other eleven attacks within our repository as a validation set for the tuning of defense hyperparameters."* While I appreciate the efforts in adding new results on the validation attacks, I am afraid this is not a scientific way of using the "validation set" in the ML context. If the authors agree that having the "validation/test split" notion is useful, I would recommend delving deeper into the correct way of utilizing the validation set, rather than simply reporting validation performance after the test set. Otherwise, "validation" may not be the appropriate term.

**Summary**

I appreciate the contribution in making ImageNet-C's corruptions differentiable so that researchers can evaluate the worst-case version of ImageNet-C (with other perturbation sets). However, I could not see a clear connection between "transforming average-case to worst-case attacks" and "benchmarking unforeseen robustness." Since ImageNet-C already introduced the concept of non-Lp perturbation sets, whatever contribution is claimed here for "unforeseen" would also apply to ImageNet-C. For example, couldn't people use the non-differentiable ImageNet-C to benchmark the "unforeseen" robustness?

Given this, I would suggest the authors explore deeper implications of making these non-Lp corruptions differentiable. I agree with the author's statement that *"It measures a different aspect of robustness that is not captured by average-case corruption robustness benchmarks or by existing adversarial robustness evaluations."* But the merit of such a "different aspect" is unclear and worth more exploration.

### Post-Discussion Update

**Actionable Feedback**
* Strengthen the logical connection between "making non-Lp perturbations differentiable" and "benchmarking the robustness of unforeseen attacks."
* Compare the proposed "worst-case benchmark" with the previous "average-case benchmark" in terms of why the new benchmark can evaluate "unforeseen robustness" better.
* Systematlize the use of "validation" and "test" sets of attacks.

---

> ### Author Response · Authors · 2023-11-21
> **Author Response**
>
> Thank you for your careful analysis of our work. We hope the following response addresses your concerns.
>
> **Defenses are not allowed to train on the test-time attacks.**
>
> >once these attacks are evaluated, they are no longer "unseen."
>
> We specify in our threat model and Section 4.2 that the defender is not allowed to train models or tune hyperparameters using the test-time attacks. This mirrors existing robustness benchmarks like ImageNet-C and is a standard way to do things. Like ImageNet-C, we also include a validation set of corruptions (in our case, these corruptions are optimizable adversarial corruptions) for tuning hyperparameters.
>
> There is a danger that the community may overfit methods to the benchmark over time, similar to concerns of overfitting on CIFAR-10 or ImageNet. However, this is not a weakness of our benchmark in particular but rather a fundamental issue that all benchmarks face.
>
> >the defense should not explicitly optimize toward these attacks.
>
> We fully agree with you. In fact, in the original submission we state in our threat model that training models or tuning hyperparameters on the test-time adversaries is forbidden. We have added additional text to the threat model section to emphasize this point (highlighted in blue).
>
> > To make the benchmark more useful, I feel it important to develop a validation/test split from these attacks
>
> We do have a validation/test split for the attacks. The eight "core" attacks are for testing only, and the eleven "extra" attacks are for validation. We have clarified this in the updated paper.
>
> >This paper includes some evaluation by adversarial training with these attacks, and the performance indeed improves. However, this is obvious and cannot provide any confidence to even newer attacks
>
> We do not adversarially train against the test-time attacks. We do include experiments with L_p adversarially trained models, which are not part of the set of test-time attacks.
>
> **Generalization outside the provided attacks.**
>
> >While this benchmark includes a wide range of empirical attacks, it is unclear if the measured robustness against these 19 now-seen attacks would generalize to robustness against other unforeseen attacks outside these 19 attacks
>
> Our benchmark contains a large number of diverse attacks, which we created by carefully designing novel strong adversaries. Methods are not allowed to train on these attacks, which means that if a method obtains high accuracy on all of our test-time attacks, it's statistically likely that the method would obtain high accuracy on new attacks created using a similar process. This is the same reason why people consider ImageNet-C scores to be meaningful.
>
> **Systematic reasons for the selection of attacks.**
>
> We do design the attacks manually. Our benchmark is partly motivated by unforeseen adversaries that could implement novel attacks to, e.g., fool an adblocker or copyright filter. It is also motivated by long tail robustness to worst-case image corruptions, including natural corruptions. These motivations guide our selection of attacks. We include digital corruptions similar to what you might see on YouTube videos trying to evade a copyright filter. We also include worst-case versions of common corruptions, like JPEG noise, snow, or fog. We have updated the paper to make these systematic reasons for our selection of attacks more clear. If we have addressed the thrust of your concerns, we kindly ask that you consider raising your score.
>
> **Additional points.**
>
> >Typos like nineteen vs eighteen, how many attacks indeed?
>
> Our benchmark includes nineteen attacks, eighteen of which are novel. We state this in the paper, but we understand that it can be confusing and have updated the text to clarify this point.

---

> ### Comment · Reviewer_PXCR · 2023-11-22
>
> Thanks for the response, and below are my remaining concerns.
>
> **Generalization beyond proposed attacks.**
>
> I am curious if there was any support for the claim *"if a method obtains high accuracy on all of our test-time attacks, it's statistically likely that the method would obtain high accuracy on new attacks created using a similar process"*?
>
> **Generalization within proposed attacks.**
>
> It seems that this paper did not rigorously evaluate the generalization from validation to test split. Instead, the notion of validation was simply tagged to the unevaluated attacks after I mentioned it.
>
> For example, it is stated that the other 11 attacks are treated as the validation split for defense tuning, but no results are reported, such as how the defense was tuned, how they performed on these 11 attacks, and how they preserved the validation robustness on the test split. It is not very convincing that the authors have actually used such a validation set, as one could easily add a "rain" or "ice" attack (as an analogy to the "snow" attack) to the validation set without doing any experiments.
>
> **Systematic reasons for the selection of attacks.**
>
> Thanks for clarifying the high-level strategy of choosing attack categories, but my main concern is how you choose the perturbation set within the same attack category. For example, why these specific digital attacks were chosen when considering digital attacks? Why the snow or fog attacks, but not rain or ice attacks, were chosen for worst-case common corruptions? These choices still look ad-hoc, meaning one could easily propose another set of attacks, not knowing which set is more suitable.

---

> ### Author Response · Authors · 2023-11-23
> **Author Response**
>
> Thank you for your quick reply. We sincerely value your time and willingness to engage in dialogue. Below, we've responded to your questions in the order they appear.
>
> **Response to Q1.**
>
> > I am curious if there was any support for the claim "if a method obtains high accuracy on all of our test-time attacks, it's statistically likely that the method would obtain high accuracy on new attacks created using a similar process"?
>
> This follows from standard statistics, e.g., the Law of Large Numbers. Since the test-time attacks are unseen and there is no overfitting to them, accuracies on them represent an empirical distribution that approximates the true distribution of accuracies on attacks sampled from a similar generating process, so a large set of attacks like the one in our benchmark can provide good assurance that one is truly making progress on the underlying problem. This is a similar reason to why people think ImageNet-C results are meaningful, or indeed why people trust that standard ImageNet test set results will generalize to new images from the same data distribution. Of course, we can never be certain that results will generalize, so the best we can do is to keep designing better evaluations. Our benchmark is a step in that direction.
>
> **Response to Q2.**
>
> > it is stated that the other 11 attacks are treated as the validation split for defense tuning, but no results are reported, such as how the defense was tuned, how they performed on these 11 attacks
>
> Our evaluations use off-the-shelf models with default hyperparameters, and all adversarially trained models in our evaluations use training-time attacks that aren't from our validation split of attacks. Therefore, we don't use the validation attacks to tune hyperparameters or perform adversarial training. We provide the validation attacks primarily to help future research on this problem. This follows the ImageNet-C paper, which provides extra corruptions for training methods or tuning hyperparameters.
>
> To show how the methods that we evaluate perform on the validation attacks, we have added full results on the 11 validation attacks across our entire set of >60 ImageNet models that we evaluate. These are in Figure 24 in the updated paper. Thank you for your suggestion.
>
> > one could easily add a "rain" or "ice" attack (as an analogy to the "snow" attack) to the validation set
>
> This would be effectively training on the test attacks by using a validation attack that is too similar to the test attacks. Our goal is to evaluate unforeseen robustness, so we require that the specific validation attacks are not too similar to the specific test attacks. A similar requirement exists for ImageNet-C; in the ImageNet-C paper, the authors specify that training on similar corruptions to the test corruptions (e.g., subtly different types of blur) is not allowed. We have updated the paper to clarify this. Thank you for your suggestion.
>
> **Response to Q3.**
>
> > For example, why these specific digital attacks were chosen when considering digital attacks? Why the snow or fog attacks, but not rain or ice attacks, were chosen for worst-case common corruptions? These choices still look ad-hoc
>
> We did not directly pick corruptions from ImageNet-C to make differentiable, so the process didn't involve selecting snow and fog instead of rain or ice. Rather, we simply brainstormed different interesting optimizable corruptions and implemented them. This was necessarily a manual process, similar to how the ImageNet-C corruptions were selected in the first place. However, we would not call it ad-hoc; we give our reasons for selecting the types of attacks we select in our previous response. Some of our attacks do overlap with ImageNet-C corruptions, like snow and fog, but others do not. We also could have easily added a rain attack, but this would have been visually quite similar to snow. If we have addressed the thrust of your concerns, we kindly ask that you consider raising your score.
>
> You do raise an important consideration, which is that some natural corruptions are easier to make differentiable than others. For example, making differentiable ice or frost would likely be quite difficult. This was a natural bias on our generation process of brainstorming different attacks; if an idea for an attack would be very hard to implement, we probably wouldn't have implemented it. This is an issue that all research on this problem has to contend with.

---

### Official Review · Reviewer_aca9 · 2023-10-30

**Soundness:** 3 good
**Presentation:** 2 fair
**Contribution:** 3 good
**Rating:** 6
**Confidence:** 4

**Summary:**

The paper addresses the gap between L-p perturbation attacks and real adversarial attacks in the image classification setting. They provide 18 novel types of attacks (19 including a previously known) to generate a more realistic adversarial dataset based on gradient methods. .

While the attack can have full access to gradients, and model weights, the defense does not have access to train-time examples to do adversarial training. Each of the attack types is a differentiable function (not known to the defense) of the image and hyperparameters (\delta) which is then constrained to a L_p bounded by \epsilon. This is then optimized by PGD to find the adversarial hyper-parameter \delta_adv.

**Strengths:**

* Proposed a benchmark of plausible adversarial attacks through differential functions where intensity is bounded by hyperparameter norm of mixing the perturbations
* Conducted evaluation across multiple image classification models to show that proposed robustness differs from L-p and L-2 robustness
* Preliminary investigation into using L2 and Lp, and generative adversarial training for improving robustness gives mixed results (comparison between L2 and L_p cannot be conclusive due to varying results as we vary \epsilon)
* Non-optimized UA2 closer to average robustness; whereas optimized UA2 matches L-p robustness empirical results
* UA2 improves with data augmentation, perceptual adversarial training, and regularization

**Weaknesses:**

* Paper has many typos which limits its readability, and proofreading would make it significantly more accessible.
* The diversity of the choice of the proposed operators is mentioned in writing, but not quantified. Perceptually several of the proposed perturbations are indistinguishable from each other (e.g. Mix, JPEG, etc), and hence justifying each addition requires some analysis of the minimal patterns required to assess the coverage of unforeseen robustness.
* Comparison with existing ImageNet perturbation datasets when used for adversarial training is lacking. Although usage of the dataset for evaluation-only is understood, a benchmark against existing adversarial datasets ImageNet-V2 (Recht et al., 2019), ImageNet-R (Hendrycks et al., 2021a), ImageNet-Sketch (Wang et al., 2019), and ObjectNet (Barbu et al., 2019) is missing.

**Questions:**

* Justification for using the term “unforeseen” needs to be made in the text. (something like functional or otherwise would be better suited)
* Not sure if this is true: “have precisely defined perturbation sets which are not dependent on the solutions found to a relaxed constrained optimization problem”. Eqn 4 is still a relaxed constrained optimization problem.

---

> ### Author Response · Authors · 2023-11-21
> **Author Response**
>
> Thank you for your careful analysis of our work. We hope the following response addresses your concerns.
>
> >Paper has many typos which limits its readability, and proofreading would make it significantly more accessible.
>
> We have conducted another proofreading pass and fixed several typos in the updated paper. Thank you for your suggestion.
>
> >The diversity of the choice of the proposed operators is mentioned in writing, but not quantified. Perceptually several of the proposed perturbations are indistinguishable from each other (e.g. Mix, JPEG, etc), and hence justifying each addition requires some analysis of the minimal patterns required to assess the coverage of unforeseen robustness.
>
> Although visually hard to distinguish at the smaller distortion sizes, our attack generation processes are highly diverse (e.g. the Mix attack functions by interpolating between two images, while the JPEG attack functions by perturbing the fourier space). The resulting variety can be seen quantitatively in Table 2 by observing differences in model performance across attacks, and more qualitatively by observing the “high distortion” images in Appendix G, which are visually varied.
>
> >Comparison with existing ImageNet perturbation datasets when used for adversarial training is lacking. Although usage of the dataset for evaluation-only is understood, a benchmark against existing adversarial datasets ImageNet-V2 (Recht et al., 2019), ImageNet-R (Hendrycks et al., 2021a), ImageNet-Sketch (Wang et al., 2019), and ObjectNet (Barbu et al., 2019) is missing.
>
> The main difference between ImageNet-UA and these works is our focus on worst-case robustness through the use of gradient-based attacks. We agree that extra comparisons would make this distinction more justified, and have added ImageNet-R and ImageNet-Sketch analysis to Appendix I in the updated paper, as well as updating the discussion in Section 5.1. As can be seen in the results (copied below for convenience), model performance on these benchmarks mirror \emph{unoptimised} versions of our attacks. Meanwhile, UA2 favors defense techniques developed for the worst-case setting (e.g. training against an $L_p$ adversary ) over defense techniques developed for these standard distribution shift benchmarks (such as data augmentation).
>
> | Model                  | UA2 (non-optimised) | ImageNet-Sketch Acc. | ImageNet-R Acc. | UA2   |
> |------------------------|---------------------|----------------------|-----------------|-------|
> | Resnet 50              | 55.2                | 24.1                 | 36.2            | 1.6   |
> | Resnet50 + AugMix      | 59.1                | 28.5                 | 41.0            | 3.5   |
> | Resnet50 + DeepAug     | 60.2                | 29.5                 | 42.2            | 3.0   |
> | Resnet50 + Mixup       | 59.9                | 26.9                 | 39.6            | 4.8   |
> | Resnet50 + L2, (ε = 5) | 43.2                | 24.2                 | 38.9            | 13.9  |
> | Resnet50 + L∞, (ε = 8/255) | 40.6           | 18.6                 | 34.8            | 10    |

---

### Official Review · Reviewer_Sze7 · 2023-10-31

**Soundness:** 1 poor
**Presentation:** 2 fair
**Contribution:** 1 poor
**Rating:** 3
**Confidence:** 4

**Summary:**

This work introduces a new benchmarking suite called ImageNet-UA. This benchmark introduces multiple different distortions to the ImageNet dataset to evaluate unforeseen adversarial accuracy. Using this new benchmark, the authors introduce their UA2 metric, which represents the mean accuracy across a core set of their attacks. They then conduct a variety of experiments to assess how UA2 changes across models and a variety of adversarial training settings.

**Strengths:**

**Originality** - Introduces a new benchmarking suite that includes attacks/distortions not present in current other robustness evaluation tools (assuming the authors plan to release this to the public)

**Quality** - Wide range of experiments, includes multiple datasets in the evaluation

**Clarity** - Images of the attack examples provide a clear picture of the outcome of the applied methods, introduction and abstract is well written

**Weaknesses:**

Below is a summary of my concerns. Specific details and suggestions can be found in the Questions section.

* Problem and goals are unclear/not well motivated
* Unclear how this work is substantially different from prior work, specifically the ImageNet-C benchmark
* Incomplete evaluation: comparison to existing benchmarks is not present, experiments done on a subset of the benchmark
* Insights are based on UA2, the use of which lacked justification

**Questions:**

**Goals, definitions, and relation to prior work**

The descriptions of the goals and motivation for this work could benefit from additional attention, specifically in how these differ from previous work. The goals and motivations that were highlighted in the abstract and intro (in italics), and my specific comments/confusions on each of them in the context of prior work are as follows:
1. _Current adversarial examples are not realistic because they use $\ell_p$ norms_ - While I agree that $\ell_p$ norms are a poor proxy for human perception, this fell flat to me because (a) the attacks introduced here do constrain to $\ell_p$ norms and (b) the ImageNet-C [1] benchmark already evaluates model robustness outside of $\ell_p$ norms.
2. _We need benchmarks against "unforeseen" adversaries_ - While the definition for what constitutes as an "unforeseen" adversary (rather than a "seen" adversary) wasn't entirely clear, it was implied that evaluation attacks should go beyond the attacks used during adversarial training and is stated that this is contrary to prior work, citing [3]. However, it is already common practice to evaluate against other attacks and to adaptive attacks that have knowledge of the defense strategy. Additionally, the evaluation in [3] includes attacks beyond PGD, so this isn't a representative example.
3. _We need benchmarks that represent worst-case adversaries_ - It is unclear why existing adversarial robustness benchmarks like AutoAttack [2] don't already represent worst-case adversaries, given that the same level of model access is assumed here and it is not demonstrated that this benchmark is more performant than other benchmarks (more details on this point later).


**Evaluation/methodology**
* Comparison with ImageNet-C are missing. Given that it seems like the corruptions in this methodology were based on ImageNet-C, it should be included. Only evaluating the UA2 of the introduced framework doesn't provide adequate support for using this benchmark over other benchmarks.
* I struggled with understanding how this benchmark is significantly different from the ImageNet-C benchmark (because of the likeness in corruptions) or from an adversarial robustness benchmark like AutoAttack (because of the likeness in optimization over the perturbation and $\ell_p$ bounding). What is the motivation behind combining the corruptions and optimization techniques?
* What is the purpose of performing the evaluation only on the "core attacks"? Specifically, it would be helpful to clarify (a) why would you want to leave out more than half of the total attacks in the framework, (b) why these attacks were selected and (c) how the core attacks are still capable of representing the performance of the entire benchmark
* The use of UA2 requires additional justification, why is the average across attacks useful (particularly if this is meant to be an ensemble method) and why isn't the individual accuracy of the attacks reported in tables 3-7?

**Presentation**
* Figure 3 does not give a clear depiction of how the attack works. Given that the perturbations are based on the corruptions, it's unclear what exactly is being optimized. Some of the arrows are confusing (e.g., images feeding into images) and it's not mentioned what the different arrow types represent. Why is the size of the perturbation not the same as the size of the input? What is $project^1_\epsilon$?
* It's not clear what conclusions are supposed to be drawn from Table 1, given that there doesn't seem to be any trend between PGD accuracy and UA2. What do the norms next to the model names represent and how are they different from the norm in the column?
* Table 2 seems to bold the model with the highest accuracy for each attack, but given that the performance of the attacks is being assessed, it seems like it would make more sense to bold the attack with the lowest accuracy for each model.
* Awkward language in some places (e.g., table 2 caption says "we plot a range of models on the pareto frontier on imagenet-UA")

Typos/errors:
- Figures 1 and 2 not referenced in text
- Tables 3 and 4 not referenced in text
- Figures and Tables are referenced out of order
- make us of -> make use of
- out suite of attacks -> our suite of attacks
- in line which -> in line with
- which novel -> which are novel
- make use use -> make use of

---

> ### Author Response · Authors · 2023-11-21
> **Author Response (1/3)**
>
> Thank you for your careful analysis of our work. We hope the following response addresses your concerns.
>
> **Clarification of the problem and goals.**
>
> We define our threat model in Section 3, including a description of the problem and goals that we consider. For example, we formally define unforeseen robustness and specify that the defender cannot train on the test distribution of adversaries. In the updated paper, we have added additional clarifications to this section, which are highlighted in blue.
>
> Our primary motivation for considering this problem is that most adversarial robustness research considers identical train-time and test-time adversaries, but in realistic scenarios adversaries commonly develop new attacks that defenders have not seen before. We give an example of this in the introduction. The current adversarial robustness literature is not developing methods to address this generalization gap. Our benchmark is the first large-scale evaluation framework that enables studying this problem, and we show in our experiments that new methods may be required to solve it.
>
> **Substantial differences with ImageNet-C.**
>
> While our benchmark may appear similar to ImageNet-C at first glance, there are in fact substantial differences that we highlight throughout the paper (e.g., see "Common corruptions" in the Related Work, the caption for Figure 2, Section 5.1, and Appendix I). These differences are as follows.
> - We use **optimized worst-case corruptions**, whereas ImageNet-C uses **unoptimized average-case corruptions**. These have very different properties, and making corruptions optimizable is nontrivial.
> - In Figure 2, we show how our optimized corruptions can be much more damaging than unoptimized corruptions.
> - **In Appendix I (Appendix G in the original submission), we conduct a quantitative comparison with ImageNet-C**, finding that our ImageNet-UA benchmark has very different properties when using optimized corruptions. In particular, worst-case optimized corruptions may require new methods to address
>
> In your review, you mention that "comparison to existing benchmarks is not present". Based on other parts of your review, we believe you may have been referring to ImageNet-C, as we are not aware of other benchmarks in the adversarial robustness literature comparable to ours. In this case, you may have missed our comparison in Appendix I (Appendix G in the original submission), which we link to in Section 5.1. We hope this comparison addresses your concerns.
>
> **Motivation for UA2 definition.**
>
> > Insights are based on UA2, the use of which lacked justification.
>
> For the motivation behind UA2, please see "Clarifications of the problem and goals." The functional form of the metric is simply an average over different attacks, mirroring the ImageNet-C metric.
>
> > why is the average across attacks useful (particularly if this is meant to be an ensemble method)
>
> We are not proposing an ensemble method like AutoAttack. Rather, we are proposing a large number of independent attacks, each of which has a different constraint. It wouldn't make sense to take the minimum accuracy across these attacks, because different constraints have different properties. For example, AutoAttack doesn't combine L_infinity and L_2 attacks together in the same ensemble, because it's understood that these constraints have different properties. Similarly, ImageNet-C doesn't take the minimum accuracy across all corruptions, because each corruption is qualitatively different and equally important. Accordingly, using the average accuracy is a natural choice that follows prior work.

---

> ### Author Response · Authors · 2023-11-21
> **Author Response (2/3)**
>
> **Responses to Questions:**
>
> **Goals, definitions, and relation to prior work**
>
> > the attacks introduced here do constrain to l_p norms and (b) the ImageNet-C benchmark already evaluates model robustness outside of l_p norms.
>
> In the adversarial robustness literature, "l_p norm attacks" refer to attacks that are constrained in *pixel-space* by an l_p norm, such as PGD, TRADES, AutoAttack, etc. The key difference between these attacks and ours is that we constrain our perturbations in a *latent space*. This allows us to use perceptible attacks with very large pixel-space l_p differences without modifying the semantics of images. E.g., one of our attacks (the only one that is not novel) is the elastic attack developed by Xiao et al. (2018). This is constrained by an l_p norm in a latent space, but would not be considered an "l_p norm attack". The same goes for most of our other attacks, including all of our core attacks.
>
> You are correct that ImageNet-C evaluates robustness outside of pixel-space l_p norms. However, as we explain in the paper ImageNet-C is an *average-case* evaluation, whereas ours uses optimized *worst-case* corruptions. There is a substantial difference between these two, comparable to the difference between perturbations optimized by the CW attack and Gaussian noise.
>
>
> > the definition for what constitutes as an "unforeseen" adversary (rather than a "seen" adversary) wasn't entirely clear
>
> We have edited Section 3 to clarify that "unforeseen adversary" refers to an adversary that is not seen at training time. In other words, we assume that there is a test set and validation set of adversaries. This follows the convention in robustness benchmarks like ImageNet-C, which do not allow training or fine-tuning hyperparameters on the test-time corruptions. Thank you for your suggestion.
>
>
> > it is already common practice to evaluate against other attacks and to adaptive attacks that have knowledge of the defense strategy
>
> We agree that prior works have studied robustness to distribution shifts in the adversaries seen at training and test time. We do not claim that we are the first to investigate this problem, and we cite published papers on this topic in the related work. We hope this demonstrates that many people find this problem interesting and worth studying. As we note in the related work, these prior works do not use a unified evaluation and only consider a small set of adversaries. Our benchmark is designed to help advance this subfield of adversarial robustness by providing a standardized evaluation with a much larger set of adversaries than explored before.
>
> Adaptive attacks are a separate concept from unforeseen attacks. The idea of adaptive attacks is to bypass particular defenses that use non-differentiability or other tricks to foil standard attacks. Adaptive attacks typically use the *same perturbation set* or constraint with a different optimizer. Unforeseen attacks are about robustness to *different perturbation sets*. This is a nuanced but important point; the difference is substantial.
>
> > It is unclear why existing adversarial robustness benchmarks like AutoAttack [2] don't already represent worst-case adversaries
>
> The RobustBench adversarial leaderboards use L_infinity and L_2 AutoAttack adversaries and allows defenses to train on these attacks. Our unforeseen robustness evaluations contain a much wider variety of attacks, all of which have non-L_p constraints, and which defenses are not allowed to train on.
>
> That said, RobustBench is a great benchmark for evaluating robustness to in-distribution L_infinity and L_2 attacks, which is still an unsolved problem. We aren't competing with RobustBench. Rather, we are considering the different and complimentary problem of unforeseen adversarial robustness, which is not addressed by AutoAttack or ImageNet-C.

---

> ### Author Response · Authors · 2023-11-21
> **Author Response (3/3)**
>
> **Evaluation/methodology**
>
> >Comparison with ImageNet-C are missing
>
> Please see Appendix I (Appendix G in the original submission).
>
> >What is the motivation behind combining the corruptions and optimization techniques?
>
> Our primary motivation is that most adversarial robustness research considers identical train-time and test-time adversaries, but in realistic scenarios adversaries commonly develop new attacks that defenders have not seen before. Our adversarial corruptions are a set of qualitatively distinct attacks that can be used for evaluating robustness to a wide variety of unseen attacks.
>
> A secondary motivation that we don't highlight as much in the paper is long tail robustness. While ImageNet-C considers average-case corruptions, one might reasonably be concerned about worst-case scenarios that might be encountered in the long tail of possible corruptions. Long tail robustness is a serious concern in safety-critical problems like self-driving, where many nines of reliability are required.
>
> >What is the purpose of performing the evaluation only on the "core attacks"?
>
> We selected the eight core attacks to make the benchmark as interesting and easy to use as possible. Specifically, they are selected based on their effectiveness and diversity, and to reduce evaluation time. We have clarified what we mean by this in the updated paper:
> - Some of the extra attacks require significantly more steps to reduce accuracy. This means they would be expensive to run or wouldn't reduce accuracy by that much.
> - Some of the extra attacks were too correlated with PGD accuracy, so we decided not to use them in the main evaluation to increase the diversity of the main evaluation.
> - The total evaluation time of the benchmark is considerable compared to ImageNet-C evaluations, so using a smaller set of curated adversaries is desirable.
>
> Note that ImageNet-C includes 4 extra corruptions for validation and hyperparameter tuning. Similarly, we treat the 11 extra attacks as a validation set of attacks.
>
> >why isn't the individual accuracy of the attacks reported in tables 3-7?
>
> Our primary metric is UA2. Tables 3-7 show the impact of various adversarial and non-adversarial robustness methods on unforeseen robustness, so we use UA2 for clarity. For many of these methods, we include the accuracy on individual attacks in Appendix D. We have included additional methods in these tables thanks to your suggestion, such as PixMix + Adversarial Training, which was previously only in the main tables.
>
> **Presentation**
> >Figure 3 does not give a clear depiction of how the attack works
>
> We have added a full description of the Wood Attack shown in Figure 3 in Appendix C. Thank you for your suggestion.
>
> >It's not clear what conclusions are supposed to be drawn from Table 1, given that there doesn't seem to be any trend between PGD accuracy and UA2. What do the norms next to the model names represent and how are they different from the norm in the column?
>
> Table 1 is meant to illustrate that UA2 measures something different from PGD, which motivates our further experiments that explore which factors can improve UA2, in which we find that new methods may be required to tackle unforeseen robustness. The norms next to model names are the attacks used for adversarial training for those models. The norm in the column is the norm used for PGD in our evaluations. We have clarified this in the updated paper. Thank you for your suggestion.
>
> >Table 2 seems to bold the model with the highest accuracy for each attack, but given that the performance of the attacks is being assessed
>
> We aren't comparing attacks to each other. The easiest way to see this is by comparison to ImageNet-C. In the ImageNet-C paper, comparing the strength of different corruptions or attacks isn't a priority. Some corruptions may indeed be more damaging than others, but all of the corruptions are concerning and worth defending against. For this reason, the ImageNet-C paper (and indeed most corruption robustness papers) focus on which methods or defenses can improve robustness to a fixed set of corruptions. Similarly, our experiment section is dedicated to investigating what factors matter for improving UA2, which is why we bold the best models and defenses rather than the strongest attacks.
>
> >Awkward language in some places (e.g., table 2 caption says "we plot a range of models on the pareto frontier on imagenet-UA")
>
> We mean that the models in Table 2 are the models that currently obtain the best performance on ImageNet-UA. We have clarified this in the updated paper. Thank you for your suggestion. If we have addressed the thrust of your concerns, we kindly ask that you consider raising your score.
>
> **Additional points**
> The benchmark will be made fully available to the public. Please see the attached supplementary material for our anonymized codebase, which is fully ready for release.
>
> We have fixed all the typos and added references to the indicated figures and tables.

---

> > ### Comment · Reviewer_Sze7 · 2023-11-22
> > **Thank you for the response, concerns about attack selection and motivation still remain**
> >
> > Hi authors, thank you for your thorough response to all the reviews and for the additional edits and inclusions to the paper. Some of my remaining concerns are as follows:
> >
> > 1. There is still a significant focus around the fact that you are evaluating unforeseen adversaries (not seen at test time). Throughout the paper it is still stated that this is a main contribution of the paper and differs from typical evaluations. However, I still feel that this misrepresents common evaluation settings. It is already common practice to evaluate on attacks that differ from the attacks used during training. For example, in [1] there is a bolded short section called "Do not only use attacks during testing that were used during training" specifically focusing on why this shouldn't be done.
> >
> > 2. It's still unclear how this benchmark should fit into the existing space of benchmarks. Is the suggestion that ImageNet-UA should be used instead of ImageNet-C? Or instead of AutoAttack? Or in addition to both of them?
> >
> > 3. The fact that the some of the attacks were left out of the core set of attacks due to the fact that they were too correlated with PGD concerns me. The conclusion from Table 1 was that $\ell_p$ robustness was distinct from UA2, were all attacks present for this evaluation? or do these two just appear distinct because attacks that tracked too closely to PGD performance were left out?
> >
> > 4. In the response it was stated that the extra 11 attacks were left out to be used as a validation set of attacks for hyperparameter tuning. However, this didn't make sense coupled with the process that you used for leaving attacks out. There was also no further information on exactly how validation and hyperparameter tuning was done on these attacks. Finally, two of my concerns from my original review remain unaddressed: more than half of the benchmark is being left out in the evaluations, why are so many attacks being left out? also, how is this still representative of the whole benchmark?
> >
> > Overall, the paper is interesting from the perspective that you are (a) designing more non $\ell_p$ attacks and (b) you introduce a methodology to optimize corruptions. But I feel like much of the paper's message and subsequent experiments is scattered and roughly focusing on this idea of unforeseen attacks and showing showing that $\ell_p$ robustness is different from non-$\ell_p$ robustness, which just did not feel like it ended up being very compelling. I think that restructuring the paper to focus more on the techniques that you introduce and how they fit into existing robustness evaluations, it would help substantially.
> >
> > [1] Nicholas Carlini, Anish Athalye, Nicolas Papernot, Wieland Brendel, Jonas Rauber, Dimitris Tsipras, Ian J. Goodfellow, Aleksander Madry, and Alexey Kurakin. On evaluating adversarial robustness.

---

> ### Author Response · Authors · 2023-11-22
> **Author Response (1/2)**
>
> Thank you for your quick reply. We sincerely value your time and willingness to engage in dialogue. Below, we've responded to your questions in the order they appear.
>
> **Response to Q1.**
>
> The term "attack" is overloaded in the literature and can refer to two very different things: (1) different optimizers and (2) different perturbation sets. There is an important distinction between the two. For example, the elastic attack of Xiao et al. (2018) [1] and AutoAttack are both distinct attacks, but Xiao et al. introduced a novel perturbation set rather than a novel optimizer, whereas AutoAttack is a stronger optimizer for standard L_p perturbation sets. Accordingly, the AutoAttack authors would not have been expected to compare to the elastic attack. We use this example to illustrate our point that there is a real difference between these two uses of the term "attack". We agree that using stronger test-time optimizers is commonplace; many adversarial robustness papers evaluate on AutoAttack. However, far fewer papers evaluate on OOD perturbation sets, and in cases where they do the number and diversity of attacks considered is fairly limited.
>
> (Note: The paper from Carlini et al. that you cite refers to using different optimizers at test-time rather than different perturbation sets. This isn't explicitly mentioned in the quoted sentence, but it's clear when reading the surrounding text.)
>
> **Response to Q2.**
>
> Our benchmark is complimentary with ImageNet-C and AutoAttack/RobustBench. It measures a different aspect of robustness that is not captured by average-case corruption robustness benchmarks or by existing adversarial robustness evaluations. We show this in the paper, finding that our benchmark may lead to the development of new methods (e.g., we find that combining corruption robustness methods with adversarial training can lead to better performance and may be a fruitful avenue for future work). We hope future work will develop networks that are robust to all three of these benchmarks.
>
> **Response to Q3.**
>
> > were all attacks present for this evaluation? or do these two just appear distinct because attacks that tracked too closely to PGD performance were left out?
>
> All the results in the main paper use our core attacks. As we mention in the paper, we selected core attacks based on effectiveness, efficiency, and lack of correlation with PGD. This was done to provide a high-quality and novel evaluation, and in Table 1 we validate that our benchmark's design in fact achieves this. We are not claiming to discover a lack of correlation with PGD in Table 1; rather, Table 1 validates the intentional design of our benchmark and simply shows that we are measuring something distinct from PGD robustness.
>
> **Key point:** Additionally, most of our extra attacks are not particularly correlated with PGD. We have added a full evaluation on the extra attacks in Figure 24 of the updated paper to show this. Of the total 11 extra attacks, only three display strong correlation with PGD (Edge, HSV, Texture). Two attacks display moderate correlation (Prison and FBM), and the remaining six display weak to negative correlation. Of these remaining six, we decided they were not effective or efficient enough to be included in the core attacks (e.g., they reduce accuracy by 10-30% instead of 60-70%). However, this does not mean that these attacks are low-quality. All of our extra attacks are visually distinct from PGD, and some have considerable novelty (e.g., our Fog attack uses a differentiable diamond-square algorithm to optimize the fog generation process; the full implementation is available in the attached supplementary ZIP file). We hope this further clarifies our process for selecting core attacks and demonstrates that our reason for doing this was simply to select the most diverse and challenging attacks for the main evaluation.

---

> ### Author Response · Authors · 2023-11-22
> **Author Response (2/2)**
>
> **Response to Q4.**
>
> > In the response it was stated that the extra 11 attacks were left out to be used as a validation set of attacks for hyperparameter tuning. However, this didn't make sense coupled with the process that you used for leaving attacks out.
>
> > why are so many attacks being left out? also, how is this still representative of the whole benchmark?
>
> Benchmarks sometimes use different distributions for validation and test sets, especially when collecting data is a lengthy or expensive process. This is the case in our benchmark, as each attack requires considerable work to design and tune (consider that many published papers have introduced only a single attack, whereas we introduce many). Our test set of core attacks can be thought of as a "hard" set, selected for being highly effective and requiring novel methods to address. The extra attacks are a different distribution of attacks (less challenging to defend against, and slightly more correlated with PGD on the whole), but we think they can still be useful for adversarial training and validation. If we have addressed the thrust of your concerns, we kindly ask that you consider raising your score.
>
> > There was also no further information on exactly how validation and hyperparameter tuning was done on these attacks
>
> Our evaluations use off-the-shelf models with default hyperparameters, so we don't use the extra attacks to tune hyperparameters in this work. We do provide the extra attacks for future work in case they are useful.
>
> In case you are referring to hyperparameter tuning on the attacks themselves, we did perform grid search hyperparameter sweeps for all 19 attacks to ensure that the low/medium/high settings were visually appropriate in the amount of information preserved, and to select default numbers of optimization steps and step sizes. This greatly improved the effectiveness and visual quality of the attacks. These hyperparameters are given in Appendix A.3.
>
> [1]: "Spatially transformed adversarial examples". Chaowei Xiao, Jun-Yan Zhu, Bo Li, Warren He, Mingyan Liu, Dawn Song. ICLR 2018

---

### Meta-Review · Area_Chair_pMyn · 2023-12-14

**Metareview:**

This paper proposes a new dataset for adversarial robustness. The gist of the contribution is that the authors make natural perturbations differentiable. All reviewers and AC appreciated the effort to extend these corruptions to become differentiable and initial experiments by the authors show that they hold promise for future research. However, unfortunately the paper suffers from serious issues that make it unpublishable at this point.

### Strengths
- All reviewers and AC believe that the set of curated attacks in this paper could be beneficial to the adversarial robustness research, and there is a non-trivial element here which is making these non-$\ell_p$ perturbations differentiable to be able to optimize over them.

- The performance on optimized perturbations is quite lower than the existing random ones in the literature, which makes these benchmarks an interesting avenue for improving model robustness. For example, it might be interesting if training on these attacks can lead to natural robustness to distribution shifts?

### Weaknesses
- There is currently little insight on how the curated attacks can benefit the field. In particular, as Reviewer PXCR has identified in the discussion, the authors lack self consistency on whether the 11 left out attacks are to be used for training models or not, making it unclear what the prescription of the authors is.

- The claim that these attacks lead to **Robustness to Unforeseen Adversarial Attacks** is not substantiated in this paper. The authors should either drop it and focus on proposing the attacks or should design new experiments to verify this hypothesis.


- Generally, the paper lacks an intuition on the  *coverage* of the 8 attacks that are suggested to be used for evaluation, and it seems that these were selected in an ad-hoc manner.

- The attacks require whitebox access to the model, which makes it unclear why being natural is important for defending against attacks or designing new ones.

- The paper currently lacks proper comparison with other existing whitebox attacks.

- There is a lack of a proper understanding of the "budget" for each of the propsoed attacks as opposed to the commonly understood $\ell_p$ attacks.

As such, after much discussion between reviewers and AC, we have decided to recommend the paper for rejection, and encourage the authors to address the shortcomings in a future submission.

**Justification For Why Not Higher Score:**

The paper suffers from serious framing issues, and in particular the main claim of the paper on Robustness to Unforeseen Adversarial Attacks is not substantiated in the paper.

**Justification For Why Not Lower Score:**

All reviewers and AC believe that the set of curated attacks in this paper could be beneficial to the adversarial robustness research, and there is a non-trivial element here which is making these non-$\ell_p$ perturbations differentiable to be able to optimize over them.

---

### Decision · Program_Chairs · 2024-01-16

Reject